# Nonlinear Sufficient Dimension Reduction with a Stochastic Neural Network

**Siqi Liang**
Purdue University
West Lafayette, IN 47906
liang257@purdue.edu

**Yan Sun**
Purdue University
West Lafayette, IN 47907
sun748@purdue.edu

**Faming Liang**[*]
Purdue University
West Lafayette, IN 47907
fmliang@purdue.edu

## Abstract

Sufficient dimension reduction is a powerful tool to extract core information hidden in the high-dimensional data and has potentially many important applications in machine learning tasks. However, the existing nonlinear sufficient dimension reduction methods often lack the scalability necessary for dealing with large-scale data. We propose a new type of stochastic neural network under a rigorous probabilistic framework and show that it can be used for sufficient dimension reduction for large-scale data. The proposed stochastic neural network is trained using an adaptive stochastic gradient Markov chain Monte Carlo algorithm, whose convergence is rigorously studied in the paper as well. Through extensive experiments on real-world classification and regression problems, we show that the proposed method compares favorably with the existing state-of-the-art sufficient dimension reduction methods and is computationally more efficient for large-scale data.

## 1 Introduction

As a supervised method, sufficient dimension reduction (SDR) aims to project the data onto a lower dimensional space so that the output is conditionally independent of the input features given the projected features. Mathematically, the problem of SDR can be described as follows. Let $\boldsymbol{Y} \in \mathbb{R}^d$ be the response variables, and let $\boldsymbol{X} = (X_1, \ldots, X_p)^T \in \mathbb{R}^p$ be the explanatory variables of dimension $p$. The goal of SDR is to find a lower-dimensional representation $\boldsymbol{Z} \in \mathbb{R}^q$, as a function of $\boldsymbol{X}$ for some $q < p$, such that

$$P(\boldsymbol{Y}|\boldsymbol{X}) = P(\boldsymbol{Y}|\boldsymbol{Z}), \quad \text{or equivalently} \quad \boldsymbol{Y} \perp\!\!\!\perp \boldsymbol{X} | \boldsymbol{Z}, \tag{1}$$

where $\perp\!\!\!\perp$ denotes conditional independence. Intuitively, the definition (1) implies that $\boldsymbol{Z}$ has extracted all the information contained in $\boldsymbol{X}$ for predicting $\boldsymbol{Y}$. In the literature, SDR has been developed under both linear and nonlinear settings.

Under the linear setting, SDR is to find a few linear combinations of $\boldsymbol{X}$ that are sufficient to describe the conditional distribution of $\boldsymbol{Y}$ given $\boldsymbol{X}$, i.e., finding a projection matrix $\boldsymbol{B} \in \mathbb{R}^{p \times q}$ such that

$$\boldsymbol{Y} \perp\!\!\!\perp \boldsymbol{X} | \boldsymbol{B}^T \boldsymbol{X}. \tag{2}$$

A more general definition for linear SDR based on $\sigma$-field can be found in [7]. Towards this goal, a variety of inverse regression methods have been proposed, see e.g., sliced inverse regression (SIR)

---

[*]To whom the correspondence should be addressed: Faming Liang.

[26], sliced average variance estimation (SAVE) [6, 8], parametric inverse regression [4], contour regression [25], and directional regression [24]. These methods require strict assumptions on the joint distribution of $(\boldsymbol{X}, \boldsymbol{Y})$ or the conditional distribution of $\boldsymbol{X}|\boldsymbol{Y}$, which limit their use in practice. To address this issue, some forward regression methods have been developed in the literature, see e.g., principal Hessian directions [27], minimum average variance estimation [45], conditional variance estimation [12], among others. These methods require minimal assumptions on the smoothness of the joint distribution $(\boldsymbol{X}, \boldsymbol{Y})$, but they do not scale well for big data problems. They can become infeasible quickly as both $p$ and $n$ increase, see [21] for more discussions on this issue.

Under the nonlinear setting, SDR is to find a nonlinear function $f(\cdot)$ such that

$$\boldsymbol{Y} \perp\!\!\!\perp \boldsymbol{X} \big| f(\boldsymbol{X}). \tag{3}$$

A general theory for nonlinear SDR has been developed in [22]. A common strategy to achieve nonlinear SDR is to apply the kernel trick to the existing linear SDR methods, where the variable $\boldsymbol{X}$ is first mapped to a high-dimensional feature space via kernels and then inverse or forward regression methods are performed. This strategy has led to a variety of methods such as kernel sliced inverse regression (KSIR) [43], kernel dimension reduction (KDR) [13, 14], manifold kernel dimension reduction (MKDR) [35], generalized sliced inverse regression (GSIR) [22], generalized sliced average variance estimator (GSAVE) [22], and least square mutual information estimation (LSMIE) [42]. A drawback shared by these methods is that they require to compute the eigenvectors or inverse of an $n \times n$ matrix. Therefore, these methods lack the scalability necessary for big data problems. Another strategy to achieve nonlinear SDR is to consider the problem under the multi-index model setting. Under this setting, the methods of forward regression such as those based on the outer product of the gradient [44, 20] have been developed, which often involve eigen-decomposition of a $p \times p$ matrix and are thus unscalable for high-dimensional problems.

Quite recently, some deep learning-based nonlinear SDR methods have been proposed in the literature, see e.g. [21, 2, 29], which are scalable for big data by training the deep neural network (DNN) with a mini-batch strategy. In [21], the authors assume that the response variable $\boldsymbol{Y}$ on the predictors $\boldsymbol{X}$ is fully captured by a regression

$$\boldsymbol{Y} = g(\boldsymbol{B}^T \boldsymbol{X}) + \boldsymbol{\epsilon}, \tag{4}$$

for an unknown function $g(\cdot)$ and a low rank parameter matrix $\boldsymbol{B}$, and they propose a two-stage approach to estimate $g(\cdot)$ and $\boldsymbol{B}$. They first estimate $g(\cdot)$ by $\tilde{g}(\cdot)$ by fitting the regression $\boldsymbol{Y} = \tilde{g}(\boldsymbol{X}) + \boldsymbol{\epsilon}$ with a DNN and initialize the estimator of $\boldsymbol{B}$ using the outer product gradient (OPG) approach [45], and then refine the estimators of $g(\cdot)$ and $\boldsymbol{B}$ by optimizing them in a joint manner. However, as pointed out by the authors, this method might not be valid unless the estimate of $g(\cdot)$ is consistent, but the consistency does not generally hold for the fully connected neural networks trained without constraints. Specifically, the universal approximation ability of the DNN can make the latent variable $\boldsymbol{Z} := \boldsymbol{B}^T \boldsymbol{X}$ unidentifiable from the DNN approximator of $g(\cdot)$; or, said differently, $\boldsymbol{Z}$ can be an arbitrary vector by tuning the size of the DNN to be sufficiently large. A similar issue happened to [2], where the authors propose to learn the latent variable $\boldsymbol{Z}$ by optimizing three DNNs to approximate the distributions $p(\boldsymbol{Z}|\boldsymbol{X})$, $p(\boldsymbol{X}|\boldsymbol{Z})$ and $p(\boldsymbol{Y}|\boldsymbol{Z})$, respectively, under the framework of variational autoencoder. Again, $\boldsymbol{Z}$ suffers from the identifiability issue due to the universal approximation ability of the DNN. In [29], the authors employ a regular DNN for sufficient dimension reduction, which works only for the case that the distribution of the response variable falls into the exponential family. How to conduct SDR with DNNs for general large-scale data remains an unresolved issue.

We address the above issue by developing a new type of stochastic neural network. The idea can be loosely described as follows. Suppose that we are able to learn a stochastic neural network, which maps $\boldsymbol{X}$ to $\boldsymbol{Y}$ via some stochastic hidden layers and possesses a layer-wise Markovian structure. Let $h$ denote the number of hidden layers, and let $\boldsymbol{Y}_1, \boldsymbol{Y}_2, \ldots, \boldsymbol{Y}_h$ denote the outputs of the respective stochastic hidden layers. By the layer-wise Markovian structure of the stochastic neural network, we can decompose the joint distribution of $(\boldsymbol{Y}, \boldsymbol{Y}_h, \boldsymbol{Y}_{h-1}, \ldots, \boldsymbol{Y}_1)$ conditioned on $\boldsymbol{X}$ as follows

$$\pi(\boldsymbol{Y}, \boldsymbol{Y}_h, \boldsymbol{Y}_{h-1}, \ldots, \boldsymbol{Y}_1 | \boldsymbol{X}) = \pi(\boldsymbol{Y}|\boldsymbol{Y}_h)\pi(\boldsymbol{Y}_h|\boldsymbol{Y}_{h-1}) \cdots \pi(\boldsymbol{Y}_1|\boldsymbol{X}), \tag{5}$$

where each conditional distribution is modeled by a linear or logistic regression (on transformed outputs of the previous layer), while the stochastic neural network still provides a good approximation to the underlying DNN under appropriate conditions on the random noise added to each stochastic layer. The layer-wise Markovian structure implies $\boldsymbol{Y} \perp\!\!\!\perp \boldsymbol{X}|\boldsymbol{Y}_h$, and the simple regression structure of $\pi(\boldsymbol{Y}|\boldsymbol{Y}_h)$ successfully gets around the identifiability issue of the latent variable $\boldsymbol{Z} := \boldsymbol{Y}_h$ that has

been suffered by some other deep learning-based methods [2, 21]. How to define and learn such a stochastic neural network will be detailed in the paper.

**Our contribution** in this paper is three-fold: (i) We propose a new type of stochastic neural network (abbreviated as "StoNet" hereafter) for sufficient dimension reduction, for which a layer-wise Markovian structure (5) is imposed on the network in training and the size of the noise added to each hidden layer is calibrated for ensuring the StoNet to provide a good approximation to the underlying DNN. (ii) We develop an adaptive stochastic gradient MCMC algorithm for training the StoNet and provides a rigorous study for its convergence under mild conditions. The training algorithm is scalable with respect to big data and it is itself of interest to statistical computing for the problems with latent variables or missing data involved. (iii) We formulate the StoNet as a composition of many simple linear/logistic regressions, making its structure more designable and interpretable. The backward imputation and forward parameter updating mechanism embedded in the proposed training algorithm enables the regression subtasks to communicate globally and update locally. As discussed later, these two features enable the StoNet to solve many important scientific problems, rather than sufficient dimension reduction, in a more convenient way than does the conventional DNN. The StoNet bridges us from linear models to deep learning.

**Other related works.** Stochastic neural networks have a long history in machine learning. Famous examples include multilayer generative models [18], restricted Boltzmann machine [19] and deep Boltzmann machine [38]. Recently, some researchers have proposed adding noise to the DNN to improve its fitting and generalization. For example, [39] proposed the dropout method to prevent the DNN from over-fitting by randomly dropping some hidden and visible units during training; [32] proposed adding gradient noise to improve training; [16, 36, 47, 40] proposed to use stochastic activation functions through adding noise to improve generalization and adversarial robustness, and [48] proposed to learn the uncertainty parameters of the stochastic activation functions along with the training of the neural network.

However, none of the existing stochastic neural networks can be used for sufficient dimension reduction. It is known that the multilayer generative models [18], restricted Boltzmann machine [19] and deep Boltzmann machine [38] can be used for dimension reduction, but under the unsupervised mode. As explained in [39], the dropout method is essentially a stochastic regularization method, where the likelihood function is penalized in network training and thus the hidden layer output of the resulting neural network does not satisfy (3). In [16], the size of the noise added to the activity function is not well calibrated and it is unclear whether the true log-likelihood function is maximized or not. The same issue happens to [32]; it is unclear whether the true log-likelihood function is maximized by the proposed training procedure. In [36], the neural network was trained by maximizing a lower bound of the log-likelihood function instead of the true log-likelihood function; therefore, its hidden layer output does not satisfy (3). In [47], the random noise added to the output of each hidden unit depends on its gradient; the mutual dependence between the gradients destroys the layer-wise Markovian structure of the neural network and thus the hidden layer output does not satisfy (3). Similarly, in [48], independent noise was added to the output of each hidden unit and, therefore, the hidden layer output satisfies neither (5) nor (3). In [40], inclusion of the support vector regression (SVR) layer to the stochastic neural network makes the hidden layer outputs mutually dependent, although the observations are mutually independent.

## 2   StoNet for Sufficient Dimension Reduction

In this section, we first define the StoNet, then justify its validity as a universal learner for the map from $X$ to $Y$ by showing that the StoNet has asymptotically the same loss function as a DNN under appropriate conditions, and further justify its use for sufficient dimension reduction.

### 2.1   The StoNet

Consider a DNN model with $h$ hidden layers. For the sake of simplicity, we assume that the same activation function $\psi$ is used for each hidden unit. By separating the feeding and activation operators

of each hidden unit, we can rewrite the DNN in the following form

$$\tilde{Y}_1 = b_1 + w_1 X,$$
$$\tilde{Y}_i = b_i + w_i \Psi(\tilde{Y}_{i-1}), \quad i = 2, 3, \ldots, h, \tag{6}$$
$$Y = b_{h+1} + w_{h+1} \Psi(\tilde{Y}_h) + e_{h+1},$$

where $e_{h+1} \sim N(0, \sigma_{h+1}^2 I_{d_{h+1}})$ is Gaussian random error; $\tilde{Y}_i, b_i \in \mathbb{R}^{d_i}$ for $i = 1, 2, \ldots, h$; $Y, b_{h+1} \in \mathbb{R}^{d_{h+1}}$; $\Psi(\tilde{Y}_{i-1}) = (\psi(\tilde{Y}_{i-1,1}), \psi(\tilde{Y}_{i-1,2}), \ldots, \psi(\tilde{Y}_{i-1,d_{i-1}}))^T$ for $i = 2, 3, \ldots, h + 1$, $\psi(\cdot)$ is the activation function, and $\tilde{Y}_{i-1,j}$ is the $j$th element of $\tilde{Y}_{i-1}$; $w_i \in \mathbb{R}^{d_i \times d_{i-1}}$ for $i = 1, 2, \ldots, h + 1$, and $d_0 = p$ denotes the dimension of $X$. For simplicity, we consider only the regression problems in (6). By replacing the third equation in (6) with a logit model, the DNN can be trivially extended to the classification problems.

The StoNet, as a probabilistic deep learning model, can be constructed by adding auxiliary noise to $\tilde{Y}_i$'s, $i = 1, 2, \ldots, h$ in (6). Mathematically, the StoNet is given by

$$Y_1 = b_1 + w_1 X + e_1,$$
$$Y_i = b_i + w_i \Psi(Y_{i-1}) + e_i, \quad i = 2, 3, \ldots, h,$$
$$Y = b_{h+1} + w_{h+1} \Psi(Y_h) + e_{h+1}, \tag{7}$$

where $Y_1, Y_2, \ldots, Y_h$ can be viewed as latent variables. Further, we assume that $e_i \sim N(0, \sigma_i^2 I_{d_i})$ for $i = 1, 2, \ldots, h, h + 1$. For classification networks, the parameter $\sigma_{h+1}^2$ plays the role of temperature for the binomial or multinomial distribution formed at the output layer, which works with $\{\sigma_1^2, \ldots, \sigma_h^2\}$ together to control the variation of the latent variables

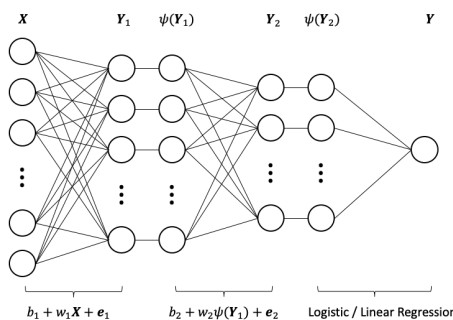

Figure 1: An illustrative plot for the structure of a StoNet with two hidden layers.

$\{Y_1, \ldots, Y_h\}$. Figure 1 depicts the architecture of the StoNet. In words, the StoNet has been formulated as a composition of many simple linear/logistic regressions, which makes its structure more designable and interpretable. Refer to Section 5 for more discussions on this issue.

## 2.2 The StoNet as an Approximator to a DNN

To show that the StoNet is a valid approximator to a DNN, i.e., asymptotically they have the same loss function, the following conditions are imposed on the model. To indicate their dependence on the training sample size $n$, we rewrite $\sigma_i$ as $\sigma_{n,i}$ for $i = 1, 2, \ldots, h + 1$. Let $\theta_i = (w_i, b_i)$, let $\theta = (\theta_1, \theta_2 \cdots, \theta_{h+1})$ denote the parameter vector of StoNet, let $d_\theta$ denote the dimension of $\theta$, and let $\Theta$ denote the space of $\theta$.

**Assumption A1** *(i) $\Theta$ is compact, i.e., $\Theta$ is contained in a $d_\theta$-ball centered at 0 with radius $r$; (ii) $\mathbb{E}(\log \pi(Y|X, \theta))^2 < \infty$ for any $\theta \in \Theta$; (iii) the activation function $\psi(\cdot)$ is $c'$-Lipschitz continuous for some constant $c'$; (iv) the network's depth $h$ and widths $d_i$'s are both allowed to increase with $n$; (v) $\sigma_{n,1} \leq \sigma_{n,2} \leq \cdots \leq \sigma_{n,h+1}$, $\sigma_{n,h+1} = O(1)$, and $d_{h+1}(\prod_{i=k+1}^{h} d_i^2)d_k \sigma_{n,k}^2 \prec \frac{1}{h}$ for any $k \in \{1, 2, \ldots, h\}$.*

Condition (i) is more or less a technical condition. As shown in Lemma S1 (in supplementary material), the proposed training algorithm for the StoNet ensures the estimates of $\theta$ to be $L_2$-upper bounded. Condition (ii) is the regularity condition for the distribution of $Y$. Condition (iii) can be satisfied by many activation functions such as *tanh*, *sigmoid* and *ReLU*. Condition (v) constrains the size of the noise added to each hidden layer such that the StoNet has asymptotically the same loss function as the DNN when the training sample size becomes large, where the factor $d_{h+1}(\prod_{i=k+1}^{h} d_i^2)d_k$ is derived in the proof of Theorem 2.1 and it can be understood as the amplification factor of the noise $e_k$ at the output layer.

Let $L : \Theta \to \mathbb{R}$ denote the loss function of the DNN as defined in (6), which is given by

$$L(\theta) = -\frac{1}{n} \sum_{i=1}^{n} \log \pi(Y^{(i)} | X^{(i)}, \theta), \tag{8}$$

where $n$ denotes the training sample size, and $i$ indexes the training samples. Theorem 2.1 shows that the StoNet and the DNN have asymptotically the same training loss function.

**Theorem 2.1** *Suppose Assumption A1 holds. Then the StoNet (7) and the neural network (6) have asymptotically the same loss function, i.e.,*

$$\sup_{\boldsymbol{\theta} \in \Theta} \left| \frac{1}{n} \sum_{i=1}^{n} \log \pi(\boldsymbol{Y}^{(i)}, \boldsymbol{Y}_{mis}^{(i)} | \boldsymbol{X}^{(i)}, \boldsymbol{\theta}) - \frac{1}{n} \sum_{i=1}^{n} \log \pi(\boldsymbol{Y}^{(i)} | \boldsymbol{X}^{(i)}, \boldsymbol{\theta}) \right| \xrightarrow{p} 0, \quad as \quad n \to \infty, \quad (9)$$

*where* $\boldsymbol{Y}_{mis} = (\boldsymbol{Y}_1, \boldsymbol{Y}_2, \ldots, \boldsymbol{Y}_h)$ *denotes the collection of all latent variables in the StoNet (7).*

Let $Q^*(\boldsymbol{\theta}) = \mathbb{E}(\log \pi(\boldsymbol{Y}|\boldsymbol{X}, \boldsymbol{\theta}))$, where the expectation is taken with respect to the joint distribution $\pi(\boldsymbol{X}, \boldsymbol{Y})$. By Assumption A1-$(i)\&(ii)$ and the law of large numbers,

$$\frac{1}{n} \sum_{i=1}^{n} \log \pi(\boldsymbol{Y}^{(i)} | \boldsymbol{X}^{(i)}, \boldsymbol{\theta}) - Q^*(\boldsymbol{\theta}) \xrightarrow{p} 0 \quad (10)$$

holds uniformly over $\Theta$. Further, we assume the following condition hold for $Q^*(\boldsymbol{\theta})$:

**Assumption A2** *(i)* $Q^*(\boldsymbol{\theta})$ *is continuous in* $\boldsymbol{\theta}$ *and uniquely maximized at* $\boldsymbol{\theta}^*$; *(ii) for any* $\epsilon > 0$, $sup_{\boldsymbol{\theta} \in \Theta \setminus B(\epsilon)} Q^*(\boldsymbol{\theta})$ *exists, where* $B(\epsilon) = \{\boldsymbol{\theta} : \|\boldsymbol{\theta} - \boldsymbol{\theta}^*\| < \epsilon\}$, *and* $\delta = Q^*(\boldsymbol{\theta}^*) - sup_{\boldsymbol{\theta} \in \Theta \setminus B(\epsilon)} Q^*(\boldsymbol{\theta}) > 0$.

Assumption A2 is more or less a technical assumption. As shown in [34] (see also [15]), for a fully connected DNN, almost all local energy minima are globally optimal if the width of one hidden layer of the DNN is no smaller than the training sample size and the network structure from this layer on is pyramidal. Similarly, [1], [11], [50], and [49] proved that the gradient-based algorithms with random initialization can converge to the global optimum provided that the width of the DNN is polynomial in training sample size. All the existing theory implies that this assumption should not be a practical concern for StoNet as long as its structure is large enough, possibly over-parameterized, such that the data can be well fitted. Further, we assume that each $\boldsymbol{\theta}$ for the DNN is unique up to loss-invariant transformations, such as reordering some hidden units and simultaneously changing the signs of some weights and biases. Such an implicit assumption has often been used in theoretical studies for neural networks, see e.g. [28] and [41] for the detail.

**Theorem 2.2** *Suppose Assumptions A1 and A2 hold, and* $\pi(\boldsymbol{Y}, \boldsymbol{Y}_{mis}|\boldsymbol{X}, \boldsymbol{\theta})$ *is continuous in* $\boldsymbol{\theta}$. *Let* $\hat{\boldsymbol{\theta}}_n = \arg\max_{\boldsymbol{\theta} \in \Theta} \{\frac{1}{n} \sum_{i=1}^{n} \log \pi(\boldsymbol{Y}^{(i)}, \boldsymbol{Y}_{mis}^{(i)}|\boldsymbol{X}^{(i)}, \boldsymbol{\theta})\}$. *Then* $\|\hat{\boldsymbol{\theta}}_n - \boldsymbol{\theta}^*\| \xrightarrow{p} 0$ *as* $n \to \infty$.

This theorem implies that the DNN (6) can be trained by training the StoNet (7), which are asymptotically equivalent as the sample size $n$ becomes large. Refer to the supplement for its proof.

### 2.3 Nonlinear Sufficient Dimension Reduction via StoNet

The joint distribution $\pi(\boldsymbol{Y}, \boldsymbol{Y}_{mis}|\boldsymbol{X}, \boldsymbol{\theta})$ for the StoNet can be factored as

$$\pi(\boldsymbol{Y}, \boldsymbol{Y}_{mis}|\boldsymbol{X}, \boldsymbol{\theta}) = \pi(\boldsymbol{Y}_1|\boldsymbol{X}, \boldsymbol{\theta}_1)[\prod_{i=2}^{h} \pi(\boldsymbol{Y}_i|\boldsymbol{Y}_{i-1}, \boldsymbol{\theta}_i)]\pi(\boldsymbol{Y}|\boldsymbol{Y}_h, \boldsymbol{\theta}_{h+1}), \quad (11)$$

based on the Markovian structure between layers of the StoNet. Therefore,

$$\pi(\boldsymbol{Y}|\boldsymbol{Y}_{mis}, \boldsymbol{X}, \boldsymbol{\theta}) = \pi(\boldsymbol{Y}|\boldsymbol{Y}_h, \boldsymbol{\theta}_{h+1}). \quad (12)$$

By Proposition 2.1 of [23], Equation (12) is equivalent to $\boldsymbol{Y} \perp\!\!\!\perp \boldsymbol{X}|\boldsymbol{Y}_h$, which coincides with the definition of nonlinear sufficient dimension reduction in (3). In summary, we have the proposition:

**Proposition 2.1** *For a well trained StoNet for the mapping* $\boldsymbol{X} \to \boldsymbol{Y}$, *the output of the last hidden layer* $\boldsymbol{Y}_h$ *satisfies SDR condition in (3).*

The proof simply follows the above arguments and the properties of the StoNet. Proposition 2.1 implies that the StoNet can be a useful and flexible tool for nonlinear SDR. However, the conventional optimization algorithm such as stochastic gradient descent (SGD) is no long applicable for training the StoNet. In the next section, we propose to train the StoNet using an adaptive stochastic gradient MCMC algorithm. At the end of the paper, we discuss how to determine the dimension of $\boldsymbol{Y}_h$ via regularization at the output layer of the StoNet.

# 3 An Adaptive Stochastic Gradient MCMC algorithm

## 3.1 Algorithm Establishment

Adaptive stochastic gradient MCMC algorithms have been developed in [10] and [9], which work under the framework of stochastic approximation MCMC [3]. Suppose that we are interested in solving the mean field equation

$$\mathbb{E}[H(\boldsymbol{Z}, \boldsymbol{\theta})] = \int H(\boldsymbol{Z}, \boldsymbol{\theta})\pi(\boldsymbol{Z}|\boldsymbol{\theta})d\boldsymbol{Z} = 0, \tag{13}$$

where $\pi(\boldsymbol{Z}|\boldsymbol{\theta})$ denotes a probability density function parameterized by $\boldsymbol{\theta}$. The adaptive stochastic gradient MCMC algorithm works by iterating between the steps: (i) *sampling*, which is to generate a Monte Carlo sample $\boldsymbol{Z}^{(k)}$ from a transition kernel that leaves $\pi(\boldsymbol{Z}|\boldsymbol{\theta}^{(k)})$ as the equilibrium distribution; and (ii) *parameter updating*, which is to update $\boldsymbol{\theta}^{(k)}$ based on the current sample $\boldsymbol{Z}^{(k)}$ in a stochastic approximation scheme. These algorithms are said "adaptive" as the transition kernel used in step (i) changes with iterations through the working estimate $\boldsymbol{\theta}^{(k)}$.

By Theorem 2.2, the StoNet can be trained by solving the equation

$$\mathbb{E}[H(\boldsymbol{Y}_{mis}, \boldsymbol{\theta})] = \int H(\boldsymbol{Y}_{mis}, \boldsymbol{\theta})\pi(\boldsymbol{Y}_{mis}|\boldsymbol{\theta}, \boldsymbol{X}, \boldsymbol{Y})d\boldsymbol{Y}_{mis} = 0, \tag{14}$$

where $H(\boldsymbol{Y}_{mis}, \boldsymbol{\theta}) = \nabla_{\boldsymbol{\theta}} \log \pi(\boldsymbol{Y}, \boldsymbol{Y}_{mis}|\boldsymbol{X}, \boldsymbol{\theta})$. Applying the adaptive stochastic gradient MCMC algorithm to (14) leads to Algorithm 1, where stochastic gradient Hamilton Monte Carlo (SGHMC) [5] is used for simulating the latent variables $\boldsymbol{Y}_{mis}$. Algorithm 1 is expected to outperform the basic algorithm by [10], where SGLD is used in the sampling step, due to the accelerated convergence of SGHMC over SGLD [33]. In Algorithm 1, we let $(\boldsymbol{Y}_0^{(s.k)}, \boldsymbol{Y}_{h+1}^{(s.k)}) = (\boldsymbol{X}^{(s)}, \boldsymbol{Y}^{(s)})$ denote a training sample $s$, and let $\boldsymbol{Y}_{mis}^{(s.k)} = (\boldsymbol{Y}_1^{(s.k)}, \ldots, \boldsymbol{Y}_h^{(s.k)})$ denote the latent variables imputed for the training sample $s$ at iteration $k$.

To make the computation for the StoNet scalable with respect to the training sample size, we train the parameter $\boldsymbol{\theta}$ with mini-batch data and then extract the SDR predictor $\boldsymbol{Y}_h$ with the full dataset; that is, we can run Algorithm 1 in two stages, namely, $\boldsymbol{\theta}$-training and SDR. In the $\boldsymbol{\theta}$-training stage, the algorithm is run with mini-batch data until convergence of $\boldsymbol{\theta}$ has been achieved; and in the SDR stage, the algorithm is run with full data for a small number of iterations. In this paper, we typically set the number of iterations/epochs of the SDR stage to 30. The proposed algorithm has the same order of computational complexity as the standard SGD algorithm, although it can be a little slower than SGD due to multiple iterations being performed at each backward sampling step.

## 3.2 Convergence Analysis of Algorithm 1

*Notations:* We let $\boldsymbol{D} = (D_1, D_2, \ldots, D_n)$ denote a dataset of $n$ observations. For StoNet, $D_i$ has included both the input and output variables of the observation. We let $\boldsymbol{Y}_{mis} = \boldsymbol{Z} = (Z_1, Z_2, \ldots, Z_n)$, where $Z_i$ is the latent variable corresponding to $D_i$, and let $f_{D_i}(z_i, \boldsymbol{\theta}) = -\log \pi(z_i|D_i, \boldsymbol{\theta})$. Let $\boldsymbol{z} = (z_1, z_2, \ldots, z_n)$ be a realization of $(Z_1, Z_2, \ldots, Z_n)$, and let $F_{\boldsymbol{D}}(\boldsymbol{Z}, \boldsymbol{\theta}) = \sum_{i=1}^n f_{D_i}(z_i, \boldsymbol{\theta})$. For simplicity, we assume $\epsilon_{k,i} = \epsilon_k$ for $i = 1, 2, \ldots, h$, and $\gamma_{k,i} = \gamma_k$ for $i = 1, 2, \ldots, h+1$.

To facilitate theoretical study, one iteration of Algorithm 1 is rewritten in the new notations as follows.

(i) (*Sampling*) Simulate the latent variable $\boldsymbol{Z}$ by setting

$$\begin{aligned}
\boldsymbol{v}^{(k+1)} &= (1 - \epsilon_{k+1}\eta)\boldsymbol{v}^{(k)} - \epsilon_{k+1}\nabla_{\boldsymbol{Z}}\hat{F}_{\boldsymbol{D}}(\boldsymbol{Z}^{(k)}, \boldsymbol{\theta}^{(k)}) + \sqrt{2\eta\epsilon_{k+1}/\beta}\boldsymbol{e}_{k+1}, \\
\boldsymbol{Z}^{(k+1)} &= \boldsymbol{Z}^{(k)} + \epsilon_{k+1}\boldsymbol{v}^{(k)},
\end{aligned} \tag{16}$$

where $\eta$ is the friction coefficient, $\beta$ is the inverse temperature, $k$ indexes the iteration, $\epsilon_{k+1}$ is the learning rate, and $\nabla_{\boldsymbol{Z}}\hat{F}_{\boldsymbol{D}}(\boldsymbol{Z}^{(k)}, \boldsymbol{\theta}^{(k)})$ is an estimate of $\nabla_{\boldsymbol{Z}}F_{\boldsymbol{D}}(\boldsymbol{Z}^{(k)}, \boldsymbol{\theta}^{(k)})$.

(ii) (*Parameter updating*) Update the parameters $\boldsymbol{\theta}$ by setting

$$\boldsymbol{\theta}^{(k+1)} = \boldsymbol{\theta}^{(k)} + \gamma_{k+1}H(\boldsymbol{Z}^{(k+1)}, \boldsymbol{\theta}^{(k)}), \tag{17}$$

where $\gamma_{k+1}$ is the step size, $H(\boldsymbol{Z}^{(k+1)}, \boldsymbol{\theta}^{(k)}) = \frac{n}{|S_k|}\sum_{i \in S_k} f_{D_i}(\boldsymbol{Z}^{(k+1)}, \boldsymbol{\theta}^{(k)})$, and $S_k$ denotes a minibatch of the full dataset.

**Algorithm 1:** An Adaptive SGHMC algorithm for training StoNet

---

**Input**: total iteration number $K$, Monte Carlo step number $t_{HMC}$, the learning rate sequence $\{\epsilon_{k,i} : t = 1, 2, \ldots, T; i = 1, 2, \ldots, h + 1\}$, and the step size sequence $\{\gamma_{k,i} : t = 1, 2, \ldots, T; i = 1, 2, \ldots, h + 1\}$;

**Initialization**: Randomly initialize the network parameters $\hat{\boldsymbol{\theta}}^{(0)} = (\hat{\theta}_1^{(0)}, \ldots, \hat{\theta}_{h+1}^{(0)})$;

**for** *k=1,2,...,K* **do**

    **STEP 0: Subsampling**: Draw a mini-batch of data and denote it by $S_k$;

    **STEP 1: Backward Sampling**

    For each observation $s \in S_k$, sample $\boldsymbol{Y}_i$'s in the order from layer $h$ to layer 1. More explicitly, we sample $\boldsymbol{Y}_i^{(s,k)}$ from the distribution

$$\pi(\boldsymbol{Y}_i^{(s,k)}|\hat{\theta}_i^{(k-1)}, \hat{\theta}_{i+1}^{(k-1)}, \boldsymbol{Y}_{i+1}^{(s,k)}, \boldsymbol{Y}_{i-1}^{(s,k)}) \propto \pi(\boldsymbol{Y}_{i+1}^{(s,k)}|\hat{\theta}_{i+1}^{(k-1)}, \boldsymbol{Y}_i^{(s,k)})\pi(\boldsymbol{Y}_i^{(s,k)}|\hat{\theta}_i^{(k-1)}, \boldsymbol{Y}_{i-1}^{(s,k)})$$

    by running SGHMC in $t_{HMC}$ steps:

    Initialize $\boldsymbol{v}_i^{(s,0)} = \boldsymbol{0}$, and initialize $\boldsymbol{Y}_i^{(s,k,0)}$ by the corresponding $\tilde{\boldsymbol{Y}}_i$ calculated in (6).

    **for** $l = 1, 2, \ldots, t_{HMC}$ **do**

        **for** $i = h, h-1, \ldots, 1$ **do**

$$\boldsymbol{v}_i^{(s,k,l)} = (1 - \epsilon_{k,i}\eta)\boldsymbol{v}_i^{(s,k,l-1)} + \epsilon_{k,i}\nabla_{\boldsymbol{Y}_i^{(s,k,l-1)}}\log\pi\left(\boldsymbol{Y}_i^{(s,k,l-1)} \mid \hat{\theta}_i^{(k-1)}, \boldsymbol{Y}_{i-1}^{(s,k,l-1)}\right)$$
$$+ \epsilon_{k,i}\nabla_{\boldsymbol{Y}_i^{(s,k,l-1)}}\log\pi\left(\boldsymbol{Y}_{i+1}^{(s,k,l-1)} \mid \hat{\theta}_{i+1}^{(k-1)}, \boldsymbol{Y}_i^{(s,k,l-1)}\right) + \sqrt{2\epsilon_{k,i}\eta}\boldsymbol{e}^{(s,k,l)},$$
$$\boldsymbol{Y}_i^{(s,k,l)} = \boldsymbol{Y}_i^{(s,k,l-1)} + \epsilon_{k,i}\boldsymbol{v}_i^{(s,k,l-1)},$$

        (15)

        where $\boldsymbol{e}^{s,k,l} \sim N(0, \boldsymbol{I}_{d_i})$, $\epsilon_{k,i}$ is the learning rate, and $\eta$ is the friction coefficient.

        **end**

    **end**

    Set $\boldsymbol{Y}_i^{(s,k)} = \boldsymbol{Y}_i^{(s,k,t_{HMC})}$ for $i = 1, 2, \ldots, h$.

    **STEP 2: Parameter Update**

    Update the estimates of $\hat{\boldsymbol{\theta}}^{(k-1)} = (\hat{\theta}_1^{(k-1)}, \hat{\theta}_2^{(k-1)}, \ldots, \hat{\theta}_{h+1}^{(k-1)})$ by

$$\hat{\theta}_i^{(k)} = \hat{\theta}_i^{(k-1)} + \gamma_{k,i}\frac{n}{|S_k|}\sum_{s \in S_k}\nabla_{\theta_i}\log\pi(Y_i^{(s,k)}|\hat{\theta}_i^{(k-1)}, Y_{i-1}^{(s,k)}), \quad i = 1, 2, \ldots, h+1,$$

    where $\gamma_{k,i}$ is the step size used for updating $\theta_i$.

**end**

---

**Theorem 3.1** *Suppose Assumptions B1-B6 (in the supplementary material) hold. If we set $\epsilon_k = C_\epsilon/(c_e + k^\alpha)$ and $\gamma_k = C_\gamma/(c_g + k^\alpha)$ for some constants $\alpha \in (0, 1)$, $C_\epsilon > 0$, $C_\gamma > 0$, $c_e \geq 0$ and $c_g \geq 0$, then there exists an iteration $k_0$ and a constant $\lambda_0 > 0$ such that for any $k > k_0$,*

$$\mathbb{E}(\|\boldsymbol{\theta}^{(k)} - \boldsymbol{\theta}^*\|^2) \leq \lambda_0\gamma_k, \tag{18}$$

*where $\boldsymbol{\theta}^*$ denotes a solution to the equation (14), and the explicit form of $\lambda_0$ is given in Theorem S1.*

Let $\mu_{D,k}$ denote the probability law of $(\boldsymbol{Z}^{(k)}, \boldsymbol{v}^{(k)})$ given the dataset $\boldsymbol{D}$, let $\pi_D$ denote the target distribution $\pi(\boldsymbol{z}|\boldsymbol{\theta}^*, \boldsymbol{D})$, let $T_k = \sum_{i=0}^{k-1}\epsilon_{i+1}$, and let $\mathcal{H}_\rho(\cdot, \cdot)$ denote a semi-metric for probability distributions. Theorem 3.2 establishes convergence of $\mu_{D,k}$.

**Theorem 3.2** *Suppose Assumptions B1-B7 (in the supplementary material) hold. Then for any $k \in \mathbb{N}$,*

$$\mathcal{W}_2(\mu_{D,T_k}, \pi_D) \leq C\sqrt{\mathcal{H}_\rho(\mu_0, \pi_D)}e^{-\mu_* T_k} + \sqrt{C_5\log(T_k)}\left(\sqrt{\tilde{C}(k)} + \left(\frac{\tilde{C}(k)}{2}\right)^{1/4}\right) + \sqrt{C_6 T_k\sum_{j=1}^{k-1}\epsilon_{j+1}^2},$$

*which can be made arbitrarily small by choosing a large enough value of $T_k$ and small enough values of $\epsilon_1$ and $\gamma_1$, provided that $\{\epsilon_k\}$ and $\{\gamma_k\}$ are set as in Theorem S1. Here $C_5$ and $C_6$ denote some constants and an explicit form of $\tilde{C}(k)$ is given in Theorem S2.*

As implied by Theorem 3.2, $\boldsymbol{Z}^{(k)}$ converges weakly to the distribution $\pi(\boldsymbol{Z}|\boldsymbol{\theta}^*, \boldsymbol{D})$ as $k \to \infty$, which ensure validity of the decomposition (11) and thus the followed SDR.

We note that our theory is very different from [10]. First, $\Theta$ is essentially assumed to be bounded in [10], while our study is under the assumption $\Theta = \mathbb{R}^{d_\theta}$. Second, for weak convergence of latent variables, only the convergence of the ergodicity average is studied in [10], while we study their convergence in 2-Wasserstein distance such that (11) holds and SDR can be further applied.

# 4 Numerical Studies

In this section, we empirically evaluate the performance of the StoNet on SDR tasks. We first compare the StoNet with an existing deep SDR method, which validates the StoNet as a SDR method. Then we compare the StoNet with some existing linear and nonlinear SDR methods on classification and regression problems. For each problem, we first apply the StoNet and the linear and nonlinear SDR methods to project the training samples onto a lower dimensional subspace, and then we train a separate classification/regression model with the projected training samples. A good SDR method is expected to extract the response information contained in the input data as much as possible. The example for multi-label classification is presented in the supplement.

## 4.1 A Validating Example for StoNet

We use the M1 example in [21] to illustrate the non-identifiability issue suffered by the deep SDR methods developed in [21] and [2]. The dataset consists of 100 independent observations. Each observation is generated from the model $\boldsymbol{Y} = \cos(\boldsymbol{X}^T\boldsymbol{b}) + \boldsymbol{\epsilon}$, where $\boldsymbol{X} \in \mathbb{R}^{20}$ follows a multivariate Gaussian distribution, $\boldsymbol{b} \in \mathbb{R}^{20}$ is a vector with the first 6 dimensions equal to $\frac{1}{\sqrt{6}}$ and the other dimensions equal to 0, and $\boldsymbol{\epsilon}$ follows a generalized Gaussian distribution $GN(0, \sqrt{1/2}, 0.5)$. We use the code [2] provided by [21] to conduct the experiment, which projects the data to one-dimensional space by working with a refinement network of structure 20-1-512-1. Let $\boldsymbol{Z}_1$ and $\boldsymbol{Z}_2$ denote two SDR vectors produced by the method in two independent runs with different initializations of network weights. We then test the independence of $\boldsymbol{Z}_1$ and $\boldsymbol{Z}_2$ using the R package RCIT [3]. The test returns a $p$-value of 0.4068, which does not reject the null hypothesis that $\boldsymbol{Z}_1$ and $\boldsymbol{Z}_2$ are independent.

We have also applied the proposed method to the same dataset, where the StoNet has a structure of 20-10-1-1 and $\tanh$ is used as the activation function. In this way, the StoNet projects the data to one-dimensional space. The SDR vectors produced in two independent runs (with different initializations of network weights) of the method are collected and tested for their dependence. The test returns a $p$-value of 0.012, which suggests that the two SDR vectors are not independent.

This example suggests that if a complicated neural network model is used to fit the function $g(\cdot)$ in (4), then the dimension reduced data do not necessarily capture the information of the original data.

## 4.2 Classification Examples

We first test the StoNet on some binary classification examples taken from [37] and [46]. The dimensions of these examples are generally low, ranging from 6 to 21. There are two steps for each example: first, we project the training samples onto a low-dimensional subspace with dimension $q = \lfloor p/2 \rfloor$ or $q = \lfloor p/4 \rfloor$ and then train a logistic regression model on the projected predictors for the binary classification task. We trained two one-hidden-layer StoNets with $\lfloor p/2 \rfloor$ and $\lfloor p/4 \rfloor$ hidden units, respectively. For comparison, four state-of-the-art non-linear SDR methods, including LSMIE, GSIR, GSAVE and KDR [4], were trained to extract nonlinear sufficient predictors. In addition, three popular linear dimension reduction methods were taken as baselines for comparison, which include SIR, SAVE and PCA [5]. The hyperparameters of these methods were determined with 5-fold

---

[2]The code is available at `https://git.art-ist.cc/daniel/NNSDR/src/branch/master`.

[3]The package is available at `https://github.com/ericstrobl/RCIT`.

[4] The code for LSMIE is available at `http://www.ms.k.u-tokyo.ac.jp/software.html#LSDR`; the code for GSIR is from Chapter 13 of [23]; the code for GSAVE is from Chapter 14 of [23]; and the code for KDR is available at `https://www.ism.ac.jp/~fukumizu/software.html`.

[5]The codes for SIR and SAVE are available in the package *sliced* downloadable at `https://joshloyal.github.io/sliced/`; and the code for PCA is available in the package *sklearn*.

cross-validation in terms of misclassification rates. Random partitioning of the dataset in cross-validation makes their results slightly different in different runs, even when the methods themselves are deterministic. Refer to the supplement for the parameter settings used in the experiments.

The results are summarized in Table 1, which reports the mean and standard deviation of the misclassification rates averaged over 20 independent trials. Table 1 shows that StoNet compares favorably to the existing linear and nonlinear dimension reduction methods.

Table 1: Mean misclassification rates on test sets (with standard deviations given in the parentheses) over 20 independent trials for some binary classification examples. In each row, the best result and those comparable to the best one (in a $t$-test at a significance level of 0.05) are highlighted in boldface.

| Datasets | q | StoNet | LSMIE | GSIR | GSAVE | KDR | SIR | SAVE | PCA |
|---|---|---|---|---|---|---|---|---|---|
| thyroid | 1 | **0.0687(.0068)** | 0.2860(.0109) | **0.0640(.0063)** | 0.0913(.0102) | 0.2847(.0110) | 0.1373(.0117) | 0.3000(.0110) | 0.3013(.0110) |
| | 2 | **0.0693(.0068)** | 0.1733(.0113) | **0.0667(.0071)** | 0.0947(.0103) | 0.2713(.0128) | 0.1373(.0130) | 0.3000(.0118) | 0.1467(.0143) |
| breastcancer | 2 | **0.2578(.0074)** | 0.2812(.0110) | 0.2772(.0091) | 0.2740(.0069) | **0.2714(.0102)** | 0.2818(.0125) | 0.2870(.0075) | 0.2857(.0129) |
| | 4 | **0.2682(.0113)** | **0.2760(.0118)** | **0.2740(.0100)** | 0.2805(.0076) | **0.2740(.0105)** | **0.2831(.0110)** | 0.2922(.0147) | **0.2766(.0097)** |
| flaresolar | 2 | **0.3236(.0040)** | 0.3770(.0177) | **0.3305(.0034)** | **0.3308(.0033)** | 0.4161(.0138) | **0.3312(.0052)** | 0.4860(.0127) | **0.3313(.0046)** |
| | 4 | **0.3239(.0043)** | 0.3346(.0043) | 0.3400(.0040) | 0.3336(.0038) | 0.3673(.0108) | **0.3328(.0049)** | 0.4302(.0133) | 0.3612(.0036) |
| heart | 3 | **0.1625(.0076)** | **0.1725(.0073)** | **0.1645(.0069)** | 0.1731(.0060) | 0.1870(.0064) | **0.1720(.0088)** | 0.1910(.0053) | 0.1920 (.0123) |
| | 6 | **0.1625(.0062)** | **0.1695(.0073)** | **0.1650(.0068)** | 0.1754(.0063) | **0.1715(.0075)** | 0.1770(.0100) | **0.1720(.0073)** | 0.1830(.0102) |
| german | 5 | **0.2368(.0050)** | 0.25(.0052) | **0.2325(.0058)** | **0.2323(.0050)** | 0.2430(.0050) | **0.2367(.0068)** | 0.2703(.0072) | 0.2777(.0070) |
| | 10 | **0.2356(.0047)** | 0.2443(.0056) | **0.2327(.0046)** | **0.2312(.0047)** | 0.2347(.0075) | **0.2360(.0068)** | 0.2447(.0062) | **0.2350(.0051)** |
| waveform | 5 | **0.1091(.0010)** | 0.1336(.0013) | 0.1140(.0015) | **0.1095(.0016)** | 0.1269(.0031) | 0.1453(.0018) | 0.1427(.0020) | 0.1486(.0013) |
| | 10 | **0.1079(.0012)** | 0.1369(.0018) | 0.1117(.0009) | **0.1070(.0013)** | 0.1254(.0030) | 0.1444(.0017) | 0.1417(.0020) | 0.1430(.0020) |

We have also tested the methods on a multi-label classification problem with a sub-MNIST dataset. Refer to Section 1 (of the supplement) for the details of the experiments. The numerical results are summarized in Table 2, which indicates the superiority of SToNet over the existing nonlinear SDR methods in both computational efficiency and prediction accuracy.

Table 2: Misclassification rates on the test set for the MNIST example, where the best misclassification rates achieved by different methods at each dimension $q$ are specified by bold face. The CPU time (in seconds) was recorded on a computer of 2.2 GHz.

| q | StoNet | LSMIE | GSIR | GSAVE | Autoencoder | PCA |
|---|---|---|---|---|---|---|
| 392 | **0.0456** | - | 0.0596 | 0.0535 | 0.1965 | 0.1002 |
| 196 | **0.0484** | - | 0.0686 | 0.0611 | 0.2268 | 0.0782 |
| 98 | **0.0503** | - | 0.0756 | 0.0696 | 0.2733 | 0.0843 |
| 49 | **0.0520** | - | 0.0816 | 0.0764 | 0.3112 | 0.0889 |
| 10 | **0.0825** | - | 0.0872 | 0.0901 | 0.4036 | 0.1644 |
| Average Time(s) | 96.18 | $> 24 hours$ | 16005.59 | 22154.11 | 1809.18 | 5.11 |

## 4.3 A Regression Example

The dataset, relative location of CT slices on axial axis [6], contains 53,500 CT images collected from 74 patients. There are 384 features extracted from the CT images, and the response is the relative location of the CT slice on the axial axis of the human body which is between 0 and 180 (where 0 denotes the top of the head and 180 the soles of the feet). Our goal is to predict the relative position of CT slices with the high-dimensional features.

Due to the large scale of the dataset and the high computation cost of the nonlinear SDR methods LSMIE, GSIR and GSAVE, we don't include them as baselines here. Similar to the previous examples, the experiment was conducted in two stages. First, we applied the dimension reduction methods to project the data onto a lower-dimensional subspace, and then trained a DNN on the projected data for making predictions. Note that for the StoNet, the dimension reduced data can be modeled by a linear regression in principle and the DNN is used here merely for fairness of comparison; while for autoencoder and PCA, the use of the DNN for modeling the dimension reduced data seems necessary for such a nonlinear regression problem. The mean squared error (MSE) and Pearson correlation were used as the evaluation metrics to assess the performance of prediction models. For this example,

---

[6]This dataset can be downloaded from UCI Machine Learning repository.

we have also trained a DNN with one hidden layer and 100 hidden units as the comparison baseline. Refer to the supplementary material for the hyperparameter settings used in the experiments.

Table 3: Mean MSE and Pearson correlation on the test sets (and their standard deviations in the parentheses) over 10 trails for the Relative location of CT slices on axial axis dataset.

| | StoNet | | Autoencoder | | PCA | |
|---|---|---|---|---|---|---|
| q | MSE | Corr | MSE | Corr | MSE | Corr |
| 192 | **0.0002(.0000)** | **0.9986(.0001)** | 0.0079(.0015) | 0.9267(.0147) | 0.0027(.0000) | 0.9755(.0001) |
| 96 | **0.0002(.0000** | **0.9985(.0001)** | 0.0106(.0024) | 0.9002(.0237) | 0.0026(.0000) | 0.9756(.0001) |
| 48 | **0.0002(.0000)** | **0.9982(.0001)** | 0.0143(.0035) | 0.8562(.0399) | 0.0034(.0000) | 0.9682(.0001) |
| 24 | **0.0002(.0000)** | **0.9980(.0001)** | 0.0185(.0033) | 0.8168(.0364) | 0.0042(.0000) | 0.9612(.0001) |
| 12 | **0.0002(.0000)** | **0.9980(.0001)** | 0.0233(.0027) | 0.7579(.0338) | 0.0053(.0000) | 0.9499(.0001) |
| 6 | **0.0002(.0000)** | **0.9980(.0001)** | 0.0304(.0024) | 0.6668(.0300) | 0.0102(.0001) | 0.9023(.0002) |
| 3 | **0.0004(.0000)** | **0.9965(.0002)** | 0.0384(.0030) | 0.5529(.0538) | 0.0209(.0001) | 0.7858(.0004) |

The results, which are summarized in Figure S2 (in the supplement) and Table 3, show that as the dimension $q$ decreases, the performance of Autoencoder degrades significantly. In contrast, StoNet can achieve stable and robust performance even when $q$ is reduced to 3. Moreover, for each value of $q$, StoNet outperforms Autoencoder and PCA significantly in both MSE and Pearson correlation.

## 5    Conclusion

In this paper, we have proposed the StoNet as a new type of stochastic neural network under the rigorous probabilistic framework and used it as a method for nonlinear SDR. The StoNet, as an approximator to neural networks, possesses a layer-wise Markovian structure and SDR can be obtained by extracting the output of its last hidden layer. The StoNet overcomes the limitations of the existing nonlinear SDR methods, such as inability in dealing with high-dimensional data and computationally inefficiency for large-scale data. We have also proposed an adaptive stochastic gradient MCMC algorithm for training the StoNet and studied its convergence theory. Extensive experimental results show that the StoNet method compares favorably with the existing state-of-the-art nonlinear SDR methods and is computationally more efficient for large-scale data.

In this paper, we study SDR with a given network structure under the assumption that the network structure has been large enough for approximating the underlying true nonlinear function. To determine the optimal depth, layer width, etc., we can combine the proposed method with a sparse deep learning method, e.g. [17] and [41]. That is, we can start with an over-parameterized neural network, employ a sparse deep learning technique to learn the network structure, and then employ the proposed method to the learned sparse DNN for the SDR task. We particularly note that the work [41] ensures consistency of the learned sparse DNN structure, which can effectively avoid the over sufficiency issue, e.g., learning a trivial relation such as identity in the components.

As an alternative way to avoid the over sufficiency issue, we can add a post sparsification step to the StoNet, i.e., applying a sparse SDR procedure (e.g., [30] and [31] with a Lasso penalty) to the output layer regression of a learnt StoNet by noting that the regression formed at each node of the StoNet is a multiple index model. In this way, the StoNet, given its universal approximation ability, provides a simple method for determining the central space of SDR for general nonlinear regression models.

The StoNet has great potentials in machine learning applications. Like other stochastic neural networks [16, 36, 47, 40], it can be used to improve generalization and adversarial robustness of the DNN. The StoNet can be easily extended to other neural network architectures such as convolutional neural network (CNN), recurrent neural network (RNN) and Long Short-Term Memory (LSTM) networks. For CNN, the randomization technique (7) can be directly applied to the fully connected layers. The same technique can be applied to appropriate layers of the RNN and LSTM as well.

## Acknowledgments

Liang's research is support in part by the NSF grants DMS-2015498 and DMS-2210819, and the NIH grant R01-GM126089.

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
