# Supplementary Material for "Nonlinear Sufficient Dimension Reduction with a Stochastic Neural Network"

**Siqi Liang**
Purdue University
West Lafayette, IN 47906
`liang257@purdue.edu`

**Yan Sun**
Purdue University
West Lafayette, IN 47907
`sun748@purdue.edu`

**Faming Liang**[*]
Purdue University
West Lafayette, IN 47907
`fmliang@purdue.edu`

This material is organized as follows. Section 1 presents more numerical Results. Section 2 proves Theorem 2.1 and Theorem 2.2. Section 3 proves Theorem 3.1 and Theorem 3.2. Section 4 presents parameter settings used in the numerical experiments.

## 1  More Numerical Results

### 1.1  A Multi-label Classification Example

We validate the effectiveness of the StoNet on the MNIST handwritten digits classification task (LeCun et al., 1998). The MNIST dataset contains 10 different classes (0 to 9) of images, including 60,000 images in the training set and 10,000 images in the test set. Each image is of size $28 \times 28$ pixels with 256 gray levels. Due to inscalability of the existing nonlinear SDR methods with respect to the sample size, we worked on a sub-training set which consisted of 20,000 images equally selected from 10 classes of the original training set.

We applied StoNet, GSIR, GSAVE, autoencoder and PCA to obtain projections onto low-dimension subspaces with the dimensions $q = 10, 49, 98, 196, 392$, and then trained a DNN on the dimension reduced data for the multi-label classification task. Note that for the StoNet, a multi-class logistic regression should work in principle for the dimension reduced data, and the DNN is used here for fairness of comparison; for some other methods such as autoencoder and PCA, the DNN seems necessary for modeling the dimension-reduced data for such a nonlinear classification problem. The StoNet consisted of one hidden layer with $q$ hidden units. All hyperparameters were determined based on 5-fold cross-validation in terms of misclassification rates. Refer to Section 4 of this material for the parameter settings used in the experiments.

The experimental results are summarized in Figure S1 and Table 2 (of the main text). For the dataset, we also trained a DNN with one hidden layer and 50 hidden units as the comparison baseline, which achieved a prediction error rate of 0.0459. The comparison shows that the StoNet outperforms GSIR, GSAVE, autoencoder and PCA in terms of misclassification rates. Moreover, StoNet is much more efficient than GSIR, GSAVE and autoencoder in computational time. It is interesting to note that when the data was projected onto a subspace with dimension 392, StoNet even outperformed the DNN in prediction accuracy. We have also tried LSMIE for this example, but lost interests finally as the method took more than 24 CPU hours on our computer.

---

[*]To whom the correspondence should be addressed: Faming Liang.

36th Conference on Neural Information Processing Systems (NeurIPS 2022).

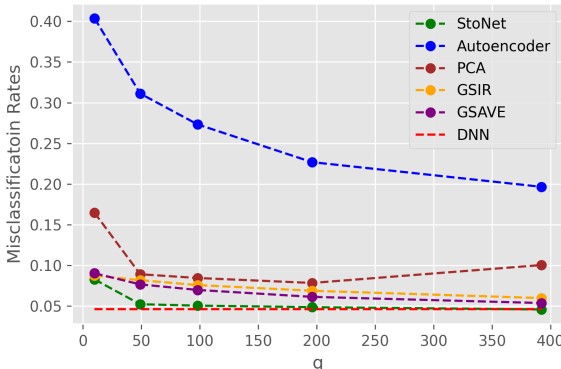

Figure S1: Misclassification rates versus dimension $q$. The red dash line represent the baseline result by training a DNN with one hidden layer and 50 hidden units on the original dataset (mistclassification rate = 0.0459).

## 1.2 A Regression Example

Refer to Figure S2 for the performance of different methods on the example.

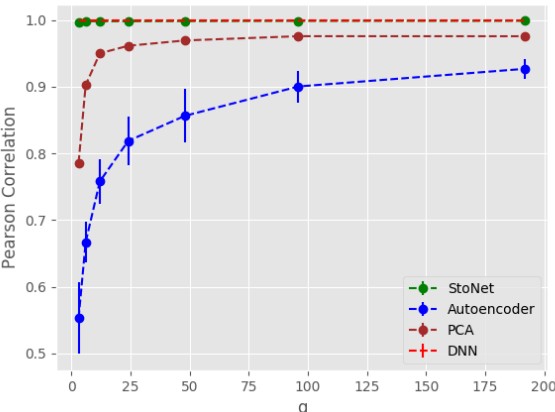

Figure S2: Pearson Correlation v.s. dimension $q$ for the regression example: The red dash line represent the baseline result by training a DNN with one hidden layer and 100 hidden units on the original dataset (Pearson correlation = 0.9987(0.0000)).

## 2 Proofs of Theorem 2.1 and Theorem 2.2

### 2.1 Proof of Theorem 2.1

PROOF: Since $\Theta$ is compact, it suffices to prove that the consistency holds for each value of $\boldsymbol{\theta}$. For simplicity of notation, we rewrite $\sigma_{n,i}$ by $\sigma_i$ in the remaining part of the proof.

Let $\boldsymbol{Y}_{mis} = (\boldsymbol{Y}_1, \boldsymbol{Y}_2, \ldots, \boldsymbol{Y}_h)$, where $\boldsymbol{Y}_i$'s are latent variables as given in Equation (6) of the main text. Let $\tilde{\boldsymbol{Y}} = (\tilde{\boldsymbol{Y}}_1, \ldots, \tilde{\boldsymbol{Y}}_h)$, where $\tilde{\boldsymbol{Y}}_i$'s are calculated by the neural network in Equation (5) of the main text. By Taylor expansion, we have

$$\log \pi(\boldsymbol{Y}, \boldsymbol{Y}_{mis}|\boldsymbol{X}, \boldsymbol{\theta}) = \log \pi(\boldsymbol{Y}, \tilde{\boldsymbol{Y}}|\boldsymbol{X}, \boldsymbol{\theta}) + \boldsymbol{\epsilon}^T \nabla_{\boldsymbol{Y}_{mis}} \log \pi(\boldsymbol{Y}, \tilde{\boldsymbol{Y}}|\boldsymbol{X}, \boldsymbol{\theta}) + O(\|\boldsymbol{\epsilon}\|^2), \quad \text{(S1)}$$

where $\epsilon = \boldsymbol{Y} - \boldsymbol{Y}_{mis} = (\epsilon_1, \epsilon_2, \ldots, \epsilon_h)$, $\log \pi(\boldsymbol{Y}, \tilde{\boldsymbol{Y}}|\boldsymbol{X}, \boldsymbol{\theta}) = \log \pi(\boldsymbol{Y}|\boldsymbol{X}, \boldsymbol{\theta})$ is the log-likelihood function of the neural network, and $\nabla_{\boldsymbol{Y}_{mis}} \log \pi(\boldsymbol{Y}, \tilde{\boldsymbol{Y}}|\boldsymbol{X}, \boldsymbol{\theta})$ is evaluated according to the joint distribution given in Equation (10) of the main text.

Consider $\nabla_{\boldsymbol{Y}_i} \log \pi(\boldsymbol{Y}, \tilde{\boldsymbol{Y}}_i|\boldsymbol{X}, \boldsymbol{\theta})$. For its single latent variable, say $Y_i^{(k)}$, the output of the hidden unit $k$ at layer $i \in \{2, \ldots, h\}$, we have

$$\nabla_{Y_i^{(k)}} \log \pi(\boldsymbol{Y}, \tilde{Y}_i^{(k)}|\boldsymbol{X}, \boldsymbol{\theta}) = \frac{1}{\sigma_{i+1}^2} \sum_{j=1}^{d_{i+1}} (Y_{i+1}^{(j)} - b_{i+1}^{(j)} - \boldsymbol{w}_{i+1}^{(j)} \psi(\tilde{\boldsymbol{Y}}_i)) w_{i+1}^{(j,k)} \psi'(\tilde{Y}_i^{(k)})$$
$$- \frac{1}{\sigma_i^2} (\tilde{Y}_i^{(k)} - b_i^{(k)} - \boldsymbol{w}_i^{(k)} \psi(\boldsymbol{Y}_{i-1})) \tag{S2}$$

where $\boldsymbol{w}_{i+1}^{(j)}$ denotes the vector of the weights from hidden unit $j$ at layer $i+1$ to the hidden units at layer $i$, and $w_{i+1}^{(j,k)}$ denotes the weight from hidden unit $j$ at layer $i+1$ to the hidden unit $k$ at hidden layer $i$. Further, by noting that $Y_{i+1}^{(j)} = b_{i+1}^{(j)} + \boldsymbol{w}_{i+1}^{(j)} \psi(\boldsymbol{Y}_i) + e_{i+1}^{(j)}$ and $Y_i^{(j)} = b_i^{(j)} + \boldsymbol{w}_i^{(j)} \psi(\boldsymbol{Y}_{i-1})$, we have

$$\nabla_{Y_i^{(k)}} \log \pi(\boldsymbol{Y}, \tilde{Y}_i^{(k)}|\boldsymbol{X}, \boldsymbol{\theta}) = \frac{1}{\sigma_{i+1}^2} \sum_{j=1}^{d_{i+1}} (e_{i+1}^{(j)} + \boldsymbol{w}_{i+1}^{(j)} (\psi(\boldsymbol{Y}_i) - \psi(\tilde{\boldsymbol{Y}}_i))) w_{i+1}^{(j,k)} \psi'(\tilde{Y}_{i,k})$$
$$- \frac{1}{\sigma_i^2} \boldsymbol{w}_i^{(k)} [\psi(\tilde{\boldsymbol{Y}}_{i-1}) - \psi(\boldsymbol{Y}_{i-1}))]. \tag{S3}$$

For layer $i = 1$, the calculation is similar, but the second term in (S3) is reduced to 0. Then by Assumption 2.1-(i)&(iv), we have

$$\left| \nabla_{Y_i^{(k)}} \log \pi \left( \boldsymbol{Y}, \tilde{Y}_i^{(k)} \mid \boldsymbol{X}, \boldsymbol{\theta} \right) \right|$$
$$\leq \begin{cases} \frac{1}{\sigma_{i+1}^2} \left\{ \sum_{j=1}^{m_{i+1}} e_{i+1}^{(j)} w_{i+1}^{(j,k)} \psi' \left( \tilde{Y}_i^{(k)} \right) + (c'r)^2 m_{i+1} \|\epsilon_i\| \right\} + \frac{1}{\sigma_i^2} c'r \|\epsilon_{i-1}\|, & \text{if } i > 1 \\ \frac{1}{\sigma_{i+1}^2} \left\{ \sum_{j=1}^{m_{i+1}} e_{i+1}^{(j)} w_{i+1}^{(j,k)} \psi' \left( \tilde{Y}_i^{(k)} \right) + (c'r)^2 m_{i+1} \|\epsilon_i\| \right\}, & \text{if } i = 1 \end{cases} \tag{S4}$$

Next, let's figure out the order of $\|\epsilon_i\|$. The $k$th component of $\epsilon_i$ is given by

$$Y_i^{(k)} - \tilde{Y}_i^{(k)} = \begin{cases} e_i^{(k)} + \boldsymbol{w}_i^{(k)} (\psi(\boldsymbol{Y}_{i-1}) - \psi(\tilde{\boldsymbol{Y}}_{i-1})), & i > 1, \\ e_i^{(k)} & i = 1. \end{cases} \tag{S5}$$

Therefore, $\|\epsilon_1\| = \|e_1\|$; and for $i = 2, 3, \ldots, h$, the following inequalities hold:

$$\|\epsilon_i\| \leq \|e_i\| + c'rd_i\|\epsilon_{i-1}\|, \quad and \quad \|\epsilon_i\|^2 \leq 2\|e_i\|^2 + 2(c'r)^2 d_i^2 \|\epsilon_{i-1}\|^2. \tag{S6}$$

Since $e_i$ and $e_{i-1}$ are independent, by summarizing (S4) and (S6), we have

$$\int \epsilon^T \nabla_{\boldsymbol{Y}_{mis}} \log \pi(\boldsymbol{Y}, \tilde{\boldsymbol{Y}}|\boldsymbol{X}, \boldsymbol{\theta}) \pi(\boldsymbol{Y}_{mis}|\boldsymbol{X}, \boldsymbol{\theta}, \boldsymbol{Y}) d\boldsymbol{Y}_{mis} \leq O\left( \sum_{k=2}^{h+1} \frac{\sigma_{k-1}^2}{\sigma_{h+1}^2} d_{h+1} (\prod_{i=k}^{h} d_i^2) d_{k-1} \right)$$
$$+ O\left( \sum_{k=2}^{h} \frac{\sigma_{k-1}^2}{\sigma_h^2} d_h (\prod_{i=k}^{h-1} d_i^2) d_{k-1} \right) + \cdots + O\left( \frac{\sigma_1^2}{\sigma_2^2} d_2 d_1 \right) = o(1), \tag{S7}$$

which, by (S1) and Assumption 2.1-$(v)$, implies the mean value

$$\mathbb{E}[\log \pi(\boldsymbol{Y}, \boldsymbol{Y}_{mis}|\boldsymbol{X}, \boldsymbol{\theta}) - \log \pi(\boldsymbol{Y}|\boldsymbol{X}, \boldsymbol{\theta})] \to 0, \quad \forall \boldsymbol{\theta} \in \Theta \tag{S8}$$

Further, it is easy to verify

$$\int |\epsilon^T \nabla_{\boldsymbol{Y}_{mis}} \log \pi(\boldsymbol{Y}, \tilde{\boldsymbol{Y}}|\boldsymbol{X}, \boldsymbol{\theta})|^2 \pi(\boldsymbol{Y}_{mis}|\boldsymbol{X}, \boldsymbol{\theta}, \boldsymbol{Y}) d\boldsymbol{Y}_{mis} < \infty, \tag{S9}$$

which, together with (S1) and (S6), implies

$$\mathbb{E}|\log \pi(\boldsymbol{Y}, \boldsymbol{Y}_{mis}|\boldsymbol{X}, \boldsymbol{\theta}) - \log \pi(\boldsymbol{Y}|\boldsymbol{X}, \boldsymbol{\theta})|^2 < \infty. \tag{S10}$$

Therefore, the weak law of large numbers (WLLN) applies, and the proof can be concluded. $\square$

## 2.2 Proof of Theorem 2.2

To prove Theorem 2.2, we first prove Lemma S1, from which Theorem 2.2 can be directly derived.

**Lemma S1** *Consider a function $Q(\boldsymbol{\theta}, \boldsymbol{X}_n)$. Suppose that the following conditions are satisfied:*

(i) *$Q(\boldsymbol{\theta}, \boldsymbol{X}_n)$ is continuous in $\boldsymbol{\theta}$ and there exists a function $Q^*(\boldsymbol{\theta})$, which is continuous in $\boldsymbol{\theta}$ and uniquely maximized at $\boldsymbol{\theta}^*$.*

(ii) *For any $\epsilon > 0$, $\sup_{\boldsymbol{\theta} \in \Theta \backslash B(\epsilon)} Q^*(\boldsymbol{\theta})$ exists, where $B(\epsilon) = \{\boldsymbol{\theta} : \|\boldsymbol{\theta} - \boldsymbol{\theta}^*\| < \epsilon\}$; Let $\delta = Q^*(\boldsymbol{\theta}^*) - \sup_{\boldsymbol{\theta} \in \Theta \backslash B(\epsilon)} Q^*(\boldsymbol{\theta})$.*

(iii) *$\sup_{\boldsymbol{\theta} \in \Theta} |Q(\boldsymbol{\theta}, \boldsymbol{X}_n) - Q^*(\boldsymbol{\theta})| \xrightarrow{p} 0$ as $n \to \infty$.*

*Let $\hat{\theta}_n = \arg\max_{\boldsymbol{\theta} \in \Theta} Q(\boldsymbol{\theta}, \boldsymbol{X}_n)$. Then $\|\hat{\boldsymbol{\theta}}_n - \boldsymbol{\theta}^*\| \xrightarrow{p} 0$.*

PROOF: Consider two events:

(a) $\sup_{\boldsymbol{\theta} \in \Theta \backslash B(\epsilon)} |Q(\boldsymbol{\theta}, \boldsymbol{X}_n) - Q^*(\boldsymbol{\theta})| < \delta/2$, and

(b) $\sup_{\boldsymbol{\theta} \in \Theta} |Q(\boldsymbol{\theta}, \boldsymbol{X}_n) - Q^*(\boldsymbol{\theta})| < \delta/2$.

From event (a), we can deduce that for any $\boldsymbol{\theta} \in \Theta \backslash B(\epsilon)$, $Q(\boldsymbol{\theta}, \boldsymbol{X}_n) < Q^*(\boldsymbol{\theta}) + \delta/2 \leq Q^*(\boldsymbol{\theta}^*) - \delta + \delta/2 \leq Q^*(\boldsymbol{\theta}^*) - \delta/2$. From event (b), we can deduce that for any $\boldsymbol{\theta} \in B(\epsilon)$, $Q(\boldsymbol{\theta}, \boldsymbol{X}_n) > Q^*(\boldsymbol{\theta}) - \delta/2$ and thus $Q(\boldsymbol{\theta}^*, \boldsymbol{X}_n) > Q^*(\boldsymbol{\theta}^*) - \delta/2$.

If both events hold simultaneously, then we must have $\hat{\boldsymbol{\theta}}_n \in B(\epsilon)$ as $n \to \infty$. By condition $(iii)$, the probability that both events hold tends to 1. Therefore, $P(\hat{\boldsymbol{\theta}}_n \in B(\epsilon)) \to 1$. $\square$

## 3 Proofs of Theorem 3.1 and Theorem 3.2

Since our goal is to obtain the SDR predictor $\boldsymbol{Y}_h$ for all observations in $\boldsymbol{D}$, we proved the convergence of Algorithm 1 for the case that the full training dataset is used at each iteration. If the algorithm is used for other purposes, say estimation of $\boldsymbol{\theta}$ only, a mini-batch of data can be used at each iteration. Extension of our proof for the mini-batch case will be discussed in Remark S2. To complete the proof, we make the following assumptions.

**Assumption B1** *The function $F_{\boldsymbol{D}}(\cdot, \cdot)$ takes nonnegative real values, and there exist constants $A, B \geq 0$, such that $|F_{\boldsymbol{D}}(\boldsymbol{0}, \boldsymbol{\theta}^*)| \leq A$, $\|\nabla_{\boldsymbol{Z}} F_{\boldsymbol{D}}(\boldsymbol{0}, \boldsymbol{\theta}^*)\| \leq B$, $\|\nabla_{\boldsymbol{\theta}} F_{\boldsymbol{D}}(\boldsymbol{0}, \boldsymbol{\theta}^*)\| \leq B$, and $\|H(\boldsymbol{0}, \boldsymbol{\theta}^*)\| \leq B$.*

**Assumption B2** *(Smoothness) $F_{\boldsymbol{D}}(\cdot, \cdot)$ is $M$-smooth and $H(\cdot, \cdot)$ is $M$-Lipschitz: there exists some constant $M > 0$ such that for any $\boldsymbol{Z}, \boldsymbol{Z}' \in \mathbb{R}^{d_z}$ and any $\boldsymbol{\theta}, \boldsymbol{\theta}' \in \Theta$,*

$$\|\nabla_{\boldsymbol{Z}} F_{\boldsymbol{D}}(\boldsymbol{Z}, \boldsymbol{\theta}) - \nabla_{\boldsymbol{Z}} F_{\boldsymbol{D}}(\boldsymbol{Z}', \boldsymbol{\theta}')\| \leq M\|\boldsymbol{Z} - \boldsymbol{Z}'\| + M\|\boldsymbol{\theta} - \boldsymbol{\theta}'\|,$$
$$\|\nabla_{\boldsymbol{\theta}} F_{\boldsymbol{D}}(\boldsymbol{Z}, \boldsymbol{\theta}) - \nabla_{\boldsymbol{\theta}} F_{\boldsymbol{D}}(\boldsymbol{Z}', \boldsymbol{\theta}')\| \leq M\|\boldsymbol{Z} - \boldsymbol{Z}'\| + M\|\boldsymbol{\theta} - \boldsymbol{\theta}'\|,$$
$$\|H(\boldsymbol{Z}, \boldsymbol{\theta}) - H(\boldsymbol{Z}', \boldsymbol{\theta}')\| \leq M\|\boldsymbol{Z} - \boldsymbol{Z}'\| + M\|\boldsymbol{\theta} - \boldsymbol{\theta}'\|.$$

**Assumption B3** *(Dissipativity) For any $\boldsymbol{\theta} \in \Theta$, the function $F_{\boldsymbol{D}}(\cdot, \boldsymbol{\theta}^*)$ is $(m, b)$-dissipative: there exist some constants $m > \frac{1}{2}$ and $b \geq 0$ such that $\langle \boldsymbol{Z}, \nabla_{\boldsymbol{Z}} F_{\boldsymbol{D}}(\boldsymbol{Z}, \boldsymbol{\theta}^*) \rangle \geq m\|\boldsymbol{Z}\|^2 - b$.*

The smoothness and dissipativity conditions are regular for studying the convergence of stochastic gradient MCMC algorithms, and they have been used in many papers such as Raginsky et al. (2017) and Gao et al. (2021). As implied by the definition of $F_{\boldsymbol{D}}(\boldsymbol{Z}, \boldsymbol{\theta})$, the values of $M$, $m$ and $b$ increase linearly with the sample size $n$. Therefore, we can impose a nonzero lower bound on $m$ to facilitate the proof of Lemma S1.

**Assumption B4** *(Gradient noise) There exists a constant $\varsigma \in [0, 1)$ such that for any $\boldsymbol{Z}$ and $\boldsymbol{\theta}$, $\mathbb{E}\|\nabla_{\boldsymbol{Z}} \hat{F}_{\boldsymbol{D}}(\boldsymbol{Z}, \boldsymbol{\theta}) - \nabla_{\boldsymbol{Z}} F_{\boldsymbol{D}}(\boldsymbol{Z}, \boldsymbol{\theta})\|^2 \leq 2\varsigma(M^2\|\boldsymbol{Z}\|^2 + M^2\|\boldsymbol{\theta} - \boldsymbol{\theta}^*\|^2 + B^2)$.*

Introduction of the extra constant $\varsigma$ facilitates our study. For the full data case, we have $\varsigma = 0$, i.e., the gradient $\nabla_{\boldsymbol{Z}} F_D(\boldsymbol{Z}, \boldsymbol{\theta})$ can be evaluated accurately.

**Assumption B5** *The step size $\{\gamma_k\}_{k \in \mathbb{N}}$ is a positive decreasing sequence such that $\gamma_k \to 0$ and $\sum_{k=1}^{\infty} \gamma_k = \infty$. In addition, let $h(\boldsymbol{\theta}) = \mathbb{E}(H(\boldsymbol{Z}, \boldsymbol{\theta}))$, then there exists $\delta > 0$ such that for any $\boldsymbol{\theta} \in \Theta$, $\langle \boldsymbol{\theta} - \boldsymbol{\theta}^*, h(\boldsymbol{\theta}) \rangle \geq \delta \|\boldsymbol{\theta} - \boldsymbol{\theta}^*\|^2$, and $\liminf_{k \to \infty} 2\delta \frac{\gamma_k}{\gamma_{k+1}} + \frac{\gamma_{k+1} - \gamma_k}{\gamma_{k+1}^2} > 0$.*

As shown by Benveniste et al. (1990) (p.244), Assumption B5 can be satisfied by setting $\gamma_k = \tilde{a}/(\tilde{b} + k^{\alpha})$ for some constants $\tilde{a} > 0$, $\tilde{b} \geq 0$, and $\alpha \in (0, 1 \wedge 2\delta\tilde{a})$. By (17), $\delta$ increases linearly with the sample size $n$. Therefore, if we set $\tilde{a} = \Omega(1/n)$ then $2\delta\tilde{a} > 1$ can be satisfied, where $\Omega(\cdot)$ denotes the order of the lower bound of a function. In this paper, we simply choose $\alpha \in (0, 1)$ by assuming that $\tilde{a}$ has been set appropriately with $2\delta\tilde{a} \geq 1$ held.

**Assumption B6** *(Solution of Poisson equation) For any $\boldsymbol{\theta} \in \Theta$, $\boldsymbol{z} \in \mathfrak{Z}$, and a function $V(\boldsymbol{z}) = 1 + \|\boldsymbol{z}\|$, there exists a function $\mu_{\boldsymbol{\theta}}$ on $\mathfrak{Z}$ that solves the Poisson equation $\mu_{\boldsymbol{\theta}}(\boldsymbol{z}) - \mathcal{T}_{\boldsymbol{\theta}}\mu_{\boldsymbol{\theta}}(\boldsymbol{z}) = H(\boldsymbol{\theta}, \boldsymbol{z}) - h(\boldsymbol{\theta})$, where $\mathcal{T}_{\boldsymbol{\theta}}$ denotes a probability transition kernel with $\mathcal{T}_{\boldsymbol{\theta}}\mu_{\boldsymbol{\theta}}(\boldsymbol{z}) = \int_{\mathfrak{Z}} \mu_{\boldsymbol{\theta}}(\boldsymbol{z}')\mathcal{T}_{\boldsymbol{\theta}}(\boldsymbol{z}, \boldsymbol{z}')d\boldsymbol{z}'$, such that*

$$H(\boldsymbol{\theta}_k, \boldsymbol{z}_{k+1}) = h(\boldsymbol{\theta}_k) + \mu_{\boldsymbol{\theta}_k}(\boldsymbol{z}_{k+1}) - \mathcal{T}_{\boldsymbol{\theta}_k}\mu_{\boldsymbol{\theta}_k}(\boldsymbol{z}_{k+1}), \quad k = 1, 2, \dots. \tag{S11}$$

*Moreover, for all $\boldsymbol{\theta}, \boldsymbol{\theta}' \in \Theta$ and $\boldsymbol{z} \in \mathfrak{Z}$, we have $\|\mu_{\boldsymbol{\theta}}(\boldsymbol{z}) - \mu_{\boldsymbol{\theta}'}(\boldsymbol{z})\| \leq \varsigma_1 \|\boldsymbol{\theta} - \boldsymbol{\theta}'\|V(\boldsymbol{z})$ and $\|\mu_{\boldsymbol{\theta}}(\boldsymbol{z})\| \leq \varsigma_2 V(\boldsymbol{z})$ for some constants $\varsigma_1 > 0$ and $\varsigma_2 > 0$.*

This assumption is also regular for studying the convergence of stochastic gradient MCMC algorithms, see e.g., Whye et al. (2016) and Deng et al. (2019). Alternatively, one can assume that the MCMC algorithms satisfy the drift condition, and then Assumption B6 can be verified, see e.g., Andrieu et al. (2005).

### 3.1 Proof of Theorem 3.1

Theorem S1 concerns the convergence of $\boldsymbol{\theta}^{(k)}$, which is a complete version of Theorem 3.1.

**Theorem S1** *(A complete version of Theorem 3.1) Suppose Assumptions B1-B6 hold. If we set $\epsilon_k = C_{\epsilon}/(c_e + k^{\alpha})$ and $\gamma_k = C_{\gamma}/(c_g + k^{\alpha})$ for some constants $\alpha \in (0, 1)$, $C_{\epsilon} > 0$, $C_{\gamma} > 0$, $c_e \geq 0$ and $c_g \geq 0$, then there exists an iteration $k_0$ and a constant $\lambda_0 > 0$ such that for any $k > k_0$,*

$$\mathbb{E}(\|\boldsymbol{\theta}^{(k)} - \boldsymbol{\theta}^*\|^2) \leq \lambda_0 \gamma_k, \tag{S12}$$

*where $\lambda_0 = \lambda_0' + 6\sqrt{6}C_{\boldsymbol{\theta}}^{\frac{1}{2}}((3M^2 + \zeta_2)C_{\boldsymbol{Z}} + 3M^2C_{\boldsymbol{\theta}} + 3B^2 + \zeta_2^2)^{\frac{1}{2}}$ for some constants $\lambda_0'$, $C_{\boldsymbol{\theta}}$ and $C_{\boldsymbol{Z}}$.*

PROOF: Our proof of Theorem 3.1 follows that of Theorem 1 in Deng et al. (2019). However, since Algorithm 1 employs SGHMC for updating $\boldsymbol{Z}^{(k)}$, which is mathematically very different from the SGLD rule employed in Deng et al. (2019), Lemma 1 of Deng et al. (2019) (uniform $L_2$ bounds of $\boldsymbol{\theta}^{(k)}$ and $\boldsymbol{Z}^{(k)}$) cannot be applied any more. In Lemma S1 below, we prove that $\mathbb{E}\|\boldsymbol{\theta}^{(k)}\|^2 \leq C_{\boldsymbol{\theta}}$, $\mathbb{E}\|\boldsymbol{v}^{(k)}\|^2 \leq C_{\boldsymbol{v}}$ and $\mathbb{E}\|\boldsymbol{Z}^{(k)}\|^2 \leq C_{\boldsymbol{Z}}$ under appropriate conditions of $\{\epsilon_k\}$ and $\{\gamma_k\}$, where $C_{\boldsymbol{\theta}}$, $C_{\boldsymbol{v}}$ and $C_{\boldsymbol{Z}}$ are appropriate constants.

Further, based on the proof of Deng et al. (2019), we can derive an explicit formula for $\lambda_0$:

$$\lambda_0 = \lambda_0' + 6\sqrt{6}C_{\boldsymbol{\theta}}^{\frac{1}{2}}((3M^2 + \zeta_2)C_{\boldsymbol{Z}} + 3M^2C_{\boldsymbol{\theta}} + 3B^2 + \zeta_2^2)^{\frac{1}{2}},$$

where $\lambda_0'$ together with $k_0$ can be derived from Lemma 3 of Deng et al. (2019) and they depend on $\delta$ and $\{\gamma_k\}$ only. The second term of $\lambda_0$ is obtained by applying the Cauchy-Schwarz inequality to bound the expectation $E\langle \boldsymbol{\theta}^{(k)} - \boldsymbol{\theta}^*, \mathcal{T}_{\boldsymbol{\theta}_{k-1}}\mu_{\boldsymbol{\theta}_{k-1}}(\boldsymbol{Z}^{(k)}) \rangle$, where $E\|\boldsymbol{\theta}^{(k)} - \boldsymbol{\theta}^*\|^2$ can be bounded according to Lemma S1 and $E\|\mathcal{T}_{\boldsymbol{\theta}^{(k-1)}}\mu_{\boldsymbol{\theta}^{(k-1)}}(\boldsymbol{Z}^{(k)})\|^2$ can be bounded according to equation (18) of Assumption B6 and the upper bound of $H(\boldsymbol{z}, \boldsymbol{\theta})$ given in (S13). $\square$

**Lemma S1** *($L_2$-bound) Suppose Assumptions 3.1-3.5 hold. If we set $\epsilon_k = C_{\epsilon}/(c_e + k^{\alpha})$ and $\gamma_k = C_{\gamma}/(c_g + k^{\alpha})$ for some constants $\alpha \in (0, 1]$, $C_{\epsilon} > 0$, $C_{\gamma} > 0$, $c_e \geq 0$ and $c_g \geq 0$, then there*

*exist constants $C_{\boldsymbol{v}}$, $C_{\boldsymbol{Z}}$ and $C_{\boldsymbol{\theta}}$ such that $\sup_{i\geq 0}\mathbb{E}\left\|\boldsymbol{v}^{(i)}\right\|^2 \leq C_{\boldsymbol{v}}$, $\sup_{i\geq 0}\mathbb{E}\left\|\boldsymbol{Z}^{(i)}\right\|^2 \leq C_{\boldsymbol{Z}}$, and $\sup_{i\geq 0}\mathbb{E}\left\|\boldsymbol{\theta}^{(i)}\right\|^2 \leq C_{\boldsymbol{\theta}}$.*

PROOF: Similar to the proof of Lemma 1 of Deng et al. (2019), we first show

$$\|\nabla_{\boldsymbol{Z}}F_{\boldsymbol{D}}(\boldsymbol{Z},\boldsymbol{\theta})\|^2 \leq 3M^2\|\boldsymbol{Z}\|^2 + 3M^2\|\boldsymbol{\theta}-\boldsymbol{\theta}^*\|^2 + 3B^2,$$
$$\|\nabla_{\boldsymbol{\theta}}F_{\boldsymbol{D}}(\boldsymbol{Z},\boldsymbol{\theta})\|^2 \leq 3M^2\|\boldsymbol{Z}\|^2 + 3M^2\|\boldsymbol{\theta}-\boldsymbol{\theta}^*\|^2 + 3B^2, \tag{S13}$$
$$\|H(\boldsymbol{Z}^{(k+1)},\boldsymbol{\theta}^{(k)})\|^2 \leq 3M^2\|\boldsymbol{Z}^{(k+1)}\|^2 + 3M^2\|\boldsymbol{\theta}^{(k)}-\boldsymbol{\theta}^*\|^2 + 3B^2.$$

By Assumption 3.1, we have $\|\nabla_{\boldsymbol{Z}}F_{\boldsymbol{D}}(\boldsymbol{0},\boldsymbol{\theta}^*)\| \leq B$, $\|H(\boldsymbol{0},\boldsymbol{\theta}^*)\| \leq B$, and $\|\nabla_{\boldsymbol{\theta}}F_{\boldsymbol{D}}(\boldsymbol{0},\boldsymbol{\theta}^*)\| \leq B$. By Assumption 3.2,

$$\|\nabla_{\boldsymbol{Z}}F_{\boldsymbol{D}}(\boldsymbol{Z},\boldsymbol{\theta})\|$$
$$\leq\|\nabla_{\boldsymbol{Z}}F_{\boldsymbol{D}}(\boldsymbol{0},\boldsymbol{\theta}^*)\| + \|\nabla_{\boldsymbol{Z}}F_{\boldsymbol{D}}(\boldsymbol{Z},\boldsymbol{\theta}^*) - \nabla_{\boldsymbol{Z}}F_{\boldsymbol{D}}(\boldsymbol{0},\boldsymbol{\theta}^*)\| + \|\nabla_{\boldsymbol{Z}}F_{\boldsymbol{D}}(\boldsymbol{Z},\boldsymbol{\theta}) - \nabla_{\boldsymbol{Z}}F_{\boldsymbol{D}}(\boldsymbol{Z},\boldsymbol{\theta}^*)\|$$
$$\leq B + M\|\boldsymbol{Z}\| + M\|\boldsymbol{\theta}-\boldsymbol{\theta}^*\|,$$
$$\|\nabla_{\boldsymbol{\theta}}F_{\boldsymbol{D}}(\boldsymbol{Z},\boldsymbol{\theta})\|$$
$$\leq\|\nabla_{\boldsymbol{\theta}}F_{\boldsymbol{D}}(\boldsymbol{0},\boldsymbol{\theta}^*)\| + \|\nabla_{\boldsymbol{\theta}}F_{\boldsymbol{D}}(\boldsymbol{Z},\boldsymbol{\theta}^*) - \nabla_{\boldsymbol{\theta}}F_{\boldsymbol{D}}(\boldsymbol{0},\boldsymbol{\theta}^*)\| + \|\nabla_{\boldsymbol{\theta}}F_{\boldsymbol{D}}(\boldsymbol{Z},\boldsymbol{\theta}) - \nabla_{\boldsymbol{Z}}F_{\boldsymbol{\theta}}(\boldsymbol{Z},\boldsymbol{\theta}^*)\|$$
$$\leq B + M\|\boldsymbol{Z}\| + M\|\boldsymbol{\theta}-\boldsymbol{\theta}^*\|,$$
$$\|H(\boldsymbol{Z}^{(k+1)},\boldsymbol{\theta}^{(k)})\|$$
$$\leq\|H(\boldsymbol{0},\boldsymbol{\theta}^*)\| + \|H(\boldsymbol{Z}^{(k+1)},\boldsymbol{\theta}^*) - H(\boldsymbol{0},\boldsymbol{\theta}^*)\| + \|H(\boldsymbol{Z}^{(k+1)},\boldsymbol{\theta}^{(k)}) - H(\boldsymbol{Z}^{(k+1)},\boldsymbol{\theta}^*)\|$$
$$\leq B + M\|\boldsymbol{Z}^{(k+1)}\| + M\|\boldsymbol{\theta}^{(k)}-\boldsymbol{\theta}^*\|.$$

Therefore, (S13) holds.

By Assumptions 3.2, 3.3 and 2.1-(i), we have

$$\langle\boldsymbol{Z},\nabla_{\boldsymbol{Z}}F_{\boldsymbol{D}}(\boldsymbol{Z},\boldsymbol{\theta})\rangle =\langle\boldsymbol{Z},\nabla_{\boldsymbol{Z}}F_{\boldsymbol{D}}(\boldsymbol{Z},\boldsymbol{\theta}^*)\rangle - \langle\boldsymbol{Z},\nabla_{\boldsymbol{Z}}F_{\boldsymbol{D}}(\boldsymbol{Z},\boldsymbol{\theta}^*) - \nabla_{\boldsymbol{Z}}F_{\boldsymbol{D}}(\boldsymbol{Z},\boldsymbol{\theta})\rangle$$
$$\geq m\|\boldsymbol{Z}\|^2 - b - \frac{1}{2}\|\boldsymbol{Z}\|^2 - \frac{1}{2}\|\nabla_{\boldsymbol{Z}}F_{\boldsymbol{D}}(\boldsymbol{Z},\boldsymbol{\theta}^*) - \nabla_{\boldsymbol{Z}}F_{\boldsymbol{D}}(\boldsymbol{Z},\boldsymbol{\theta})\|^2$$
$$\geq (m - \frac{1}{2})\|\boldsymbol{Z}\|^2 - b - \frac{1}{2}M^2\|\boldsymbol{\theta}-\boldsymbol{\theta}^*\|^2$$
$$\geq m_0\|\boldsymbol{Z}\|^2 - b - \frac{1}{2}M^2\|\boldsymbol{\theta}-\boldsymbol{\theta}^*\|^2,$$

where the constants $m_0 = m - \frac{1}{2} > 0$, Then, similar to the proof of Lemma 2 in Raginsky et al. (2017), we have

$$F_{\boldsymbol{D}}(\boldsymbol{Z},\boldsymbol{\theta}) = F_{\boldsymbol{D}}(\boldsymbol{0},\boldsymbol{\theta}^*) + \int_0^1 \langle\boldsymbol{Z},\nabla_{\boldsymbol{Z}}F_{\boldsymbol{D}}(t\boldsymbol{Z},\boldsymbol{\theta}^* + t(\boldsymbol{\theta}-\boldsymbol{\theta}^*))\rangle + \langle\boldsymbol{\theta}-\boldsymbol{\theta}^*,\nabla_{\boldsymbol{\theta}}F_{\boldsymbol{D}}(t\boldsymbol{Z},\boldsymbol{\theta}^* + t(\boldsymbol{\theta}-\boldsymbol{\theta}^*))\rangle dt$$

$$\leq A + \int_0^1 \|\boldsymbol{Z}\|\|\nabla_{\boldsymbol{Z}}F_{\boldsymbol{D}}(t\boldsymbol{Z},\boldsymbol{\theta}^* + t(\boldsymbol{\theta}-\boldsymbol{\theta}^*))\|dt + \int_0^1 \|\boldsymbol{\theta}-\boldsymbol{\theta}^*\|\|\nabla_{\boldsymbol{\theta}}F_{\boldsymbol{D}}(t\boldsymbol{Z},\boldsymbol{\theta}^* + t(\boldsymbol{\theta}-\boldsymbol{\theta}^*))\|dt$$

$$\leq A + \|\boldsymbol{Z}\|\int_0^1 tM\|\boldsymbol{Z}\| + tM\|\boldsymbol{\theta}-\boldsymbol{\theta}^*\| + Bdt + \|\boldsymbol{\theta}-\boldsymbol{\theta}^*\|\int_0^1 tM\|\boldsymbol{Z}\| + tM\|\boldsymbol{\theta}-\boldsymbol{\theta}^*\| + Bdt$$

$$\leq A + M\|\boldsymbol{Z}\|^2 + M\|\boldsymbol{\theta}-\boldsymbol{\theta}^*\|^2 + \frac{B}{2}\|\boldsymbol{Z}\|^2 + \frac{B}{2}\|\boldsymbol{\theta}-\boldsymbol{\theta}^*\|^2 + B$$

$$\leq M_0\|\boldsymbol{Z}\|^2 + A_0 + M_0\|\boldsymbol{\theta}-\boldsymbol{\theta}^*\|^2,$$

where the constants $M_0 = M + \frac{B}{2}$, and $A_0 = A + B$. Then, similar to Gao et al. (2021), for $0 < \lambda < \min\{\frac{1}{4}, \frac{m_0}{2M_0+\eta^2/2}\}$, Assumption 3.3 gives us

$$\langle\boldsymbol{Z},\nabla_{\boldsymbol{Z}}F_{\boldsymbol{D}}(\boldsymbol{Z},\boldsymbol{\theta})\rangle \geq m_0\|\boldsymbol{Z}\|^2 - b - \frac{1}{2}M^2\|\boldsymbol{\theta}-\boldsymbol{\theta}^*\|^2 \geq \lambda(2M_0 + \frac{\eta^2}{2})\|\boldsymbol{Z}\|^2 - b - \frac{1}{2}M^2\|\boldsymbol{\theta}-\boldsymbol{\theta}^*\|^2$$
$$\geq 2\lambda(F_{\boldsymbol{D}}(\boldsymbol{Z},\boldsymbol{\theta}) + \frac{\eta^2}{4}\|\boldsymbol{Z}\|^2) - \frac{A_1}{\beta} - (\frac{1}{2}M^2 + 2\lambda M_0)\|\boldsymbol{\theta}-\boldsymbol{\theta}^*\|^2,$$
$$\tag{S14}$$

where the constant $A_1 = \beta(2\lambda A_0 + b) > 0$.

For $\boldsymbol{Z}^{(k)}$ and $\boldsymbol{v}^{(k)}$, we have

$$\mathbb{E}\|\boldsymbol{Z}^{(k+1)}\|^2 = \mathbb{E}\|\boldsymbol{Z}^{(k)}\|^2 + \epsilon_{k+1}^2\mathbb{E}\|\boldsymbol{v}^{(k)}\|^2 + 2\epsilon_{k+1}\mathbb{E}\langle\boldsymbol{Z}^{(k)},\boldsymbol{v}^{(k)}\rangle, \tag{S15}$$

$$
\begin{aligned}
\mathbb{E}\|\boldsymbol{v}^{(k+1)}\|^2 &= \mathbb{E}\|(1-\epsilon_{k+1}\eta)\boldsymbol{v}^{(k)} - \epsilon_{k+1}\nabla_{\boldsymbol{Z}}\hat{F}_{\boldsymbol{D}}(\boldsymbol{Z}^{(k)},\boldsymbol{\theta}^{(k)})\|^2 + 2\epsilon_{k+1}\eta\beta^{-1}\mathbb{E}\|\boldsymbol{e}^{(k+1)}\|^2 \\
&\quad + \sqrt{8\epsilon_{k+1}\eta\beta^{-1}}\mathbb{E}\langle(1-\epsilon_{k+1}\eta)\boldsymbol{v}^{(k)} - \epsilon_{k+1}\nabla_{\boldsymbol{Z}}F_{\boldsymbol{D}}(\boldsymbol{Z}^{(k)},\boldsymbol{\theta}^{(k)}),\boldsymbol{e}_{k+1}\rangle \\
&= \mathbb{E}\|(1-\epsilon_{k+1}\eta)\boldsymbol{v}^{(k)} - \epsilon_{k+1}\nabla_{\boldsymbol{Z}}\hat{F}_{\boldsymbol{D}}(\boldsymbol{Z}^{(k)},\boldsymbol{\theta}^{(k)})\|^2 + 2\epsilon_{k+1}\eta\beta^{-1}d_z \\
&= \mathbb{E}\|(1-\epsilon_{k+1}\eta)\boldsymbol{v}^{(k)} - \epsilon_{k+1}\nabla_{\boldsymbol{Z}}F_{\boldsymbol{D}}(\boldsymbol{Z}^{(k)},\boldsymbol{\theta}^{(k)})\|^2 + 2\epsilon_{k+1}\eta\beta^{-1}d_z \\
&\quad + \epsilon_{k+1}^2\mathbb{E}\|\nabla_{\boldsymbol{Z}}F_{\boldsymbol{D}}(\boldsymbol{Z}^{(k)},\boldsymbol{\theta}^{(k)}) - \nabla_{\boldsymbol{Z}}\hat{F}_{\boldsymbol{D}}(\boldsymbol{Z}^{(k)},\boldsymbol{\theta}^{(k)})\|^2 \\
&\leq \mathbb{E}\|\boldsymbol{v}^{(k)}\|^2 + (\epsilon_{k+1}^2\eta^2 - 2\epsilon_{k+1}\eta)\mathbb{E}\|\boldsymbol{v}^{(k)}\|^2 - 2\epsilon_{k+1}(1-\epsilon_{k+1}\eta)\mathbb{E}\langle\boldsymbol{v}^{(k)},\nabla_{\boldsymbol{Z}}F_{\boldsymbol{D}}(\boldsymbol{Z}^{(k)},\boldsymbol{\theta}^{(k)})\rangle \\
&\quad + \epsilon_{k+1}^2\mathbb{E}\|\nabla_{\boldsymbol{Z}}F_{\boldsymbol{D}}(\boldsymbol{Z}^{(k)},\boldsymbol{\theta}^{(k)})\|^2 + 2\varsigma\epsilon_{k+1}^2(M^2\mathbb{E}\|\boldsymbol{Z}^{(k)}\|^2 + M^2\mathbb{E}\|\boldsymbol{\theta}^{(k)} - \boldsymbol{\theta}^*\|^2 + B^2) + 2\epsilon_{k+1}\eta\beta^{-1}d_z.
\end{aligned}
\tag{S16}
$$

Therefore, we have

$$
\begin{aligned}
&\mathbb{E}\|\boldsymbol{Z}^{(k+1)} + \eta^{-1}\boldsymbol{v}^{(k+1)}\|^2 = \mathbb{E}\|\boldsymbol{Z}^{(k)} + \eta^{-1}\boldsymbol{v}^{(k)} - \epsilon_{k+1}\eta^{-1}\nabla_{\boldsymbol{Z}}\hat{F}_{\boldsymbol{D}}(\boldsymbol{Z}^{(k)},\boldsymbol{\theta}^{(k)}) + \sqrt{2\epsilon_{k+1}\beta^{-1}\eta^{-1}}\boldsymbol{e}^{(k)}\|^2 \\
&= \mathbb{E}\|\boldsymbol{Z}^{(k)} + \eta^{-1}\boldsymbol{v}^{(k)} - \epsilon_{k+1}\eta^{-1}\nabla_{\boldsymbol{Z}}F_{\boldsymbol{D}}(\boldsymbol{Z}^{(k)},\boldsymbol{\theta}^{(k)})\|^2 + 2\epsilon_{k+1}\beta^{-1}\eta^{-1}d_z \\
&\quad + \epsilon_{k+1}^2\mathbb{E}\|\nabla_{\boldsymbol{Z}}F_{\boldsymbol{D}}(\boldsymbol{Z}^{(k)},\boldsymbol{\theta}^{(k)}) - \nabla_{\boldsymbol{Z}}\hat{F}_{\boldsymbol{D}}(\boldsymbol{Z}^{(k)},\boldsymbol{\theta}^{(k)})\|^2 \\
&\leq \mathbb{E}\|\boldsymbol{Z}^{(k)} + \eta^{-1}\boldsymbol{v}^{(k)}\|^2 - 2\epsilon_{k+1}\eta^{-1}\mathbb{E}\langle\boldsymbol{Z}^{(k)},\nabla_{\boldsymbol{Z}}F_{\boldsymbol{D}}(\boldsymbol{Z}^{(k)},\boldsymbol{\theta}^{(k)})\rangle \\
&\quad - 2\epsilon_{k+1}\eta^{-2}\langle\boldsymbol{v}^{(k)},\nabla_{\boldsymbol{Z}}F_{\boldsymbol{D}}(\boldsymbol{Z}^{(k)},\boldsymbol{\theta}^{(k)})\rangle + \epsilon_{k+1}^2\eta^{-2}\|\nabla_{\boldsymbol{Z}}F_{\boldsymbol{D}}(\boldsymbol{Z}^{(k)},\boldsymbol{\theta}^{(k)})\|^2 \\
&\quad + 2\epsilon_{k+1}\beta^{-1}\eta^{-1}d_z + 2\varsigma\epsilon_{k+1}^2(M^2\mathbb{E}\|\boldsymbol{Z}^{(k)}\|^2 + M^2\mathbb{E}\|\boldsymbol{\theta}^{(k)} - \boldsymbol{\theta}^*\|^2 + B^2).
\end{aligned}
\tag{S17}
$$

Similarly, for $\boldsymbol{\theta}^{(k+1)}$, we have

$$
\begin{aligned}
&\mathbb{E}\|\boldsymbol{\theta}^{(k+1)} - \boldsymbol{\theta}^*\|^2 \\
&= \mathbb{E}\|\boldsymbol{\theta}^{(k)} - \boldsymbol{\theta}^*\|^2 - 2\gamma_{k+1}\mathbb{E}\langle\boldsymbol{\theta}^{(k)} - \boldsymbol{\theta}^*, H(\boldsymbol{Z}^{(k+1)},\boldsymbol{\theta}^{(k)})\rangle + \gamma_{k+1}^2\mathbb{E}\|H(\boldsymbol{Z}^{(k+1)},\boldsymbol{\theta}^{(k)})\|.
\end{aligned}
$$

Recall that $h(\boldsymbol{\theta}) = \mathbb{E}(H(\boldsymbol{Z},\boldsymbol{\theta}))$, we have

$$
\begin{aligned}
\mathbb{E}\langle\boldsymbol{\theta}^{(k)} - \boldsymbol{\theta}^*, H(\boldsymbol{Z}^{(k+1)},\boldsymbol{\theta}^{(k)})\rangle &= \mathbb{E}\langle\boldsymbol{\theta}^{(k)} - \boldsymbol{\theta}^*, H(\boldsymbol{Z}^{(k+1)},\boldsymbol{\theta}^{(k)}) - h(\boldsymbol{\theta})\rangle + \mathbb{E}\langle\boldsymbol{\theta}^{(k)} - \boldsymbol{\theta}^*, h(\boldsymbol{\theta})\rangle \\
&= \mathbb{E}\langle\boldsymbol{\theta}^{(k)} - \boldsymbol{\theta}^*, h(\boldsymbol{\theta})\rangle \geq \delta\mathbb{E}\|\boldsymbol{\theta}^{(k)} - \boldsymbol{\theta}^*\|^2.
\end{aligned}
$$

Then we have

$$\mathbb{E}\|\boldsymbol{\theta}^{(k+1)} - \boldsymbol{\theta}^*\|^2 \leq (1 - 2\gamma_{k+1}\delta)\mathbb{E}\|\boldsymbol{\theta}^{(k)} - \boldsymbol{\theta}^*\|^2 + \gamma_{k+1}^2\mathbb{E}\|H(\boldsymbol{Z}^{(k+1)},\boldsymbol{\theta}^{(k)})\|^2. \tag{S18}$$

For $F_{\boldsymbol{D}}(\boldsymbol{Z}^{(k)},\boldsymbol{\theta}^{(k)})$, we have

$$
\begin{aligned}
&F_{\boldsymbol{D}}(\boldsymbol{Z}^{(k+1)},\boldsymbol{\theta}^{(k+1)}) - F_{\boldsymbol{D}}(\boldsymbol{Z}^{(k)},\boldsymbol{\theta}^{(k)}) \\
&= F_{\boldsymbol{D}}(\boldsymbol{Z}^{(k+1)},\boldsymbol{\theta}^{(k+1)}) - F_{\boldsymbol{D}}(\boldsymbol{Z}^{(k+1)},\boldsymbol{\theta}^{(k)}) + F_{\boldsymbol{D}}(\boldsymbol{Z}^{(k+1)},\boldsymbol{\theta}^{(k)}) - F_{\boldsymbol{D}}(\boldsymbol{Z}^{(k)},\boldsymbol{\theta}^{(k)}) \\
&= \int_0^1 \langle\nabla_{\boldsymbol{\theta}}F_{\boldsymbol{D}}(\boldsymbol{Z}^{(k+1)},\boldsymbol{\theta}^{(k)} + t\gamma_{k+1}H(\boldsymbol{Z}^{(k+1)},\boldsymbol{\theta}^{(k)})),\gamma_{k+1}H(\boldsymbol{Z}^{(k+1)},\boldsymbol{\theta}^{(k)})\rangle dt \\
&\quad + \int_0^1 \langle\nabla_{\boldsymbol{Z}}F_{\boldsymbol{D}}(\boldsymbol{Z}^{(k)} + t\epsilon_{k+1}\boldsymbol{v}^{(k)},\boldsymbol{\theta}^{(k)}),\epsilon_{k+1}\boldsymbol{v}^{(k)}\rangle dt.
\end{aligned}
$$

Then, by Assumption 3.2,

$$
|F_{\boldsymbol{D}}(\boldsymbol{Z}^{(k+1)}, \boldsymbol{\theta}^{(k+1)}) - F_{\boldsymbol{D}}(\boldsymbol{Z}^{(k)}, \boldsymbol{\theta}^{(k)})|
$$

$$
\leq |\langle \nabla_{\boldsymbol{\theta}} F_{\boldsymbol{D}}(\boldsymbol{Z}^{(k+1)}, \boldsymbol{\theta}^{(k)}), \gamma_{k+1} H(\boldsymbol{Z}^{(k+1)}, \boldsymbol{\theta}^{(k)}) \rangle + \langle \nabla_{\boldsymbol{Z}} F_{\boldsymbol{D}}(\boldsymbol{Z}^{(k)}, \boldsymbol{\theta}^{(k)}), \epsilon_{k+1} \boldsymbol{v}^{(k)} \rangle|
$$

$$
+ \int_0^1 \|\nabla_{\boldsymbol{\theta}} F_{\boldsymbol{D}}(\boldsymbol{Z}^{(k+1)}, \boldsymbol{\theta}^{(k)} + t\gamma_{k+1} H(\boldsymbol{Z}^{(k+1)}, \boldsymbol{\theta}^{(k)})) - \nabla_{\boldsymbol{\theta}} F_{\boldsymbol{D}}(\boldsymbol{Z}^{(k+1)}, \boldsymbol{\theta}^{(k)})\| \|\gamma_{k+1} H(\boldsymbol{Z}^{(k+1)}, \boldsymbol{\theta}^{(k)})\| dt
$$

$$
+ \int_0^1 \|\nabla_{\boldsymbol{Z}} F_{\boldsymbol{D}}(\boldsymbol{Z}^{(k)} + t\epsilon_{k+1} \boldsymbol{v}^{(k)}, \boldsymbol{\theta}^{(k)}) - \nabla_{\boldsymbol{Z}} F_{\boldsymbol{D}}(\boldsymbol{Z}^{(k)}, \boldsymbol{\theta}^{(k)})\| \|\epsilon_{k+1} \boldsymbol{v}^{(k)}\| dt
$$

$$
\leq |\langle \nabla_{\boldsymbol{\theta}} F_{\boldsymbol{D}}(\boldsymbol{Z}^{(k+1)}, \boldsymbol{\theta}^{(k)}), \gamma_{k+1} H(\boldsymbol{Z}^{(k+1)}, \boldsymbol{\theta}^{(k)}) \rangle + \langle \nabla_{\boldsymbol{Z}} F_{\boldsymbol{D}}(\boldsymbol{Z}^{(k)}, \boldsymbol{\theta}^{(k)}), \epsilon_{k+1} \boldsymbol{v}^{(k)} \rangle|
$$

$$
+ \frac{1}{2} M \gamma_{k+1}^2 \|H(\boldsymbol{Z}^{(k+1)}, \boldsymbol{\theta}^{(k)}\|^2 + \frac{1}{2} M \epsilon_{k+1}^2 \|\boldsymbol{v}^{(k)}\|^2,
$$

which implies

$$
\mathbb{E} F_{\boldsymbol{D}}(\boldsymbol{Z}^{(k+1)}, \boldsymbol{\theta}^{(k+1)})
$$

$$
\leq \mathbb{E} F_{\boldsymbol{D}}(\boldsymbol{Z}^{(k)}, \boldsymbol{\theta}^{(k)}) + (\frac{1}{2} M \gamma_{k+1}^2 + \frac{1}{2} \gamma_{k+1}) \mathbb{E} \|H(\boldsymbol{Z}^{(k+1)}, \boldsymbol{\theta}^{(k)}\|^2 + \frac{1}{2} \gamma_{k+1} \mathbb{E} \|\nabla_{\boldsymbol{\theta}} F_{\boldsymbol{D}}(\boldsymbol{Z}^{(k+1)}, \boldsymbol{\theta}^{(k)}\|^2
$$

$$
+ \frac{1}{2} M \epsilon_{k+1}^2 \mathbb{E} \|\boldsymbol{v}^{(k)}\|^2 + \epsilon_{k+1} \mathbb{E} \langle \nabla_{\boldsymbol{Z}} F_{\boldsymbol{D}}(\boldsymbol{Z}^{(k)}, \boldsymbol{\theta}^{(k)}), \boldsymbol{v}^{(k)} \rangle.
$$

$$
\text{(S19)}
$$

Now, let's consider

$$
L(k) = \mathbb{E} \left[ F_{\boldsymbol{D}}(\boldsymbol{Z}^{(k)}, \boldsymbol{\theta}^{(k)}) + \frac{3M^2 + \lambda\eta + G}{2\delta} \|\boldsymbol{\theta}^{(k)} - \boldsymbol{\theta}^*\|^2 + \frac{1}{4} \eta^2 (\|\boldsymbol{Z}^{(k)} + \eta^{-1} \boldsymbol{v}^{(k)}\|^2 + \|\eta^{-1} \boldsymbol{v}^{(k)}\|^2 - \lambda \|\boldsymbol{Z}^{(k)}\|^2) \right],
$$

where $G$ is a constant and it will be defined later. Note that for our model, $F_{\boldsymbol{D}}(\boldsymbol{Z}, \boldsymbol{\theta}) \geq 0$. Then it is easy to see that

$$
L(k) \geq \max\{\frac{3M^2 + \lambda\eta + G}{2\delta} \mathbb{E} \|\boldsymbol{\theta}^{(k)} - \boldsymbol{\theta}^*\|^2, \frac{1}{8}(1 - 2\lambda)\eta^2 \mathbb{E} \|\boldsymbol{Z}^{(k)}\|^2, \frac{1}{4}(1 - 2\lambda) \mathbb{E} \|\boldsymbol{v}^{(k)}\|^2\}. \quad \text{(S20)}
$$

We only need to provide uniform bound for $L(k)$. To complete this goal, we first study the relationship between $L(k+1)$ and $L(k)$:

$$
L(k+1) - L(k) \leq (\frac{1}{2} M \gamma_{k+1}^2 + \frac{1}{2} \gamma_{k+1}) \mathbb{E} \|H(\boldsymbol{Z}^{(k+1)}, \boldsymbol{\theta}^{(k)}\|^2 + \frac{1}{2} \gamma_{k+1} \mathbb{E} \|\nabla_{\boldsymbol{\theta}} F_{\boldsymbol{D}}(\boldsymbol{Z}^{(k+1)}, \boldsymbol{\theta}^{(k)}\|^2
$$

$$
+ \frac{1}{2} M \epsilon_{k+1}^2 \mathbb{E} \|\boldsymbol{v}^{(k)}\|^2 + \epsilon_{k+1} \mathbb{E} \langle \nabla_{\boldsymbol{Z}} F_{\boldsymbol{D}}(\boldsymbol{Z}^{(k)}, \boldsymbol{\theta}^{(k)}), \boldsymbol{v}^{(k)} \rangle
$$

$$
- (3M^2 + \lambda\eta + G)\gamma_{k+1} \mathbb{E} \|\boldsymbol{\theta}^{(k)} - \boldsymbol{\theta}^*\|^2 + \frac{(3M^2 + \lambda\eta + G)\gamma_{k+1}^2}{2\delta} \mathbb{E} \|H(\boldsymbol{Z}^{(k+1)}, \boldsymbol{\theta}^{(k)}\|^2
$$

$$
- \frac{1}{2} \epsilon_{k+1} \eta \mathbb{E} \langle \boldsymbol{Z}^{(k)}, \nabla_{\boldsymbol{Z}} F_{\boldsymbol{D}}(\boldsymbol{Z}^{(k)}, \boldsymbol{\theta}^{(k)}) \rangle - \frac{1}{2} \epsilon_{k+1} \mathbb{E} \langle \boldsymbol{v}^{(k)}, \nabla_{\boldsymbol{Z}} F_{\boldsymbol{D}}(\boldsymbol{Z}^{(k)}, \boldsymbol{\theta}^{(k)}) \rangle + \frac{1}{4} \epsilon_{k+1}^2 \mathbb{E} \|\nabla_{\boldsymbol{Z}} F_{\boldsymbol{D}}(\boldsymbol{Z}^{(k)}, \boldsymbol{\theta}^{(k)})\|^2
$$

$$
+ \frac{1}{2} \epsilon_{k+1} \beta^{-1} \eta d_z + \frac{1}{2} \varsigma \eta^2 \epsilon_{k+1}^2 (M^2 \mathbb{E} \|\boldsymbol{Z}^{(k)}\|^2 + M^2 \mathbb{E} \|\boldsymbol{\theta}^{(k)} - \boldsymbol{\theta}^*\|^2 + B^2)
$$

$$
+ \frac{1}{4}(\epsilon_{k+1}^2 \eta^2 - 2\epsilon_{k+1} \eta) \mathbb{E} \|\boldsymbol{v}^{(k)}\|^2 - \frac{1}{2} \epsilon_{k+1}(1 - \epsilon_{k+1} \eta) \mathbb{E} \langle \boldsymbol{v}^{(k)}, \nabla_{\boldsymbol{Z}} F_{\boldsymbol{D}}(\boldsymbol{Z}^{(k)}, \boldsymbol{\theta}^{(k)}) \rangle
$$

$$
+ \frac{1}{4} \epsilon_{k+1}^2 \mathbb{E} \|\nabla_{\boldsymbol{Z}} F_{\boldsymbol{D}}(\boldsymbol{Z}^{(k)}, \boldsymbol{\theta}^{(k)})\|^2 + \frac{1}{2} \epsilon_{k+1} \eta \beta^{-1} d_z + \frac{1}{2} \varsigma \epsilon_{k+1}^2 (M^2 \mathbb{E} \|\boldsymbol{Z}^{(k)}\|^2 + M^2 \mathbb{E} \|\boldsymbol{\theta}^{(k)} - \boldsymbol{\theta}^*\|^2 + B^2)
$$

$$
- \frac{1}{4} \lambda \eta^2 \epsilon_{k+1}^2 \mathbb{E} \|\boldsymbol{v}^{(k)}\|^2 - \frac{1}{2} \lambda \eta^2 \epsilon_{k+1} \mathbb{E} \langle \boldsymbol{Z}^{(k)}, \boldsymbol{v}^{(k)} \rangle
$$

$$
\leq (\frac{1}{2} M \gamma_{k+1}^2 + \gamma_{k+1} + \frac{(3M^2 + \lambda\eta + G)\gamma_{k+1}^2}{2\delta})(3M^2 (2\mathbb{E} \|\boldsymbol{Z}^{(k)}\|^2 + 2\epsilon_{k+1}^2 \mathbb{E} \|\boldsymbol{v}^{(k)}\|^2) + 3M^2 \mathbb{E} \|\boldsymbol{\theta}^{(k)} - \boldsymbol{\theta}^*\|^2 + 3B^2)
$$

$$
+ (-\frac{1}{2} \eta \epsilon_{k+1} + (\frac{1}{2} M + \frac{1}{4} \eta^2 - \frac{1}{4} \lambda \eta^2) \epsilon_{k+1}^2) \mathbb{E} \|\boldsymbol{v}^{(k)}\|^2 + \frac{1}{2} \eta \epsilon_{k+1}^2 \mathbb{E} \langle \boldsymbol{v}^{(k)}, \nabla_{\boldsymbol{Z}} F_{\boldsymbol{D}}(\boldsymbol{Z}^{(k)}, \boldsymbol{\theta}^{(k)}) \rangle
$$

$$
(-(3M^2 + \lambda\eta + G)\gamma_{k+1} + (\frac{1}{2} \varsigma M^2 \eta^2 + \frac{1}{2} \varsigma M^2) \epsilon_{k+1}^2) \mathbb{E} \|\boldsymbol{\theta}^{(k)} - \boldsymbol{\theta}^*\|^2 - \frac{1}{2} \epsilon_{k+1} \eta \mathbb{E} \langle \boldsymbol{Z}^{(k)}, \nabla_{\boldsymbol{Z}} F_{\boldsymbol{D}}(\boldsymbol{Z}^{(k)}, \boldsymbol{\theta}^{(k)}) \rangle
$$

$$
+ \frac{1}{2} \epsilon_{k+1}^2 (3M^2 \mathbb{E} \|\boldsymbol{Z}^{(k)}\|^2 + 3M^2 \mathbb{E} \|\boldsymbol{\theta}^{(k)} - \boldsymbol{\theta}^*\|^2 + 3B^2) + (\frac{1}{2} \varsigma M^2 \eta^2 + \frac{1}{2} \varsigma M^2) \epsilon_{k+1}^2 \mathbb{E} \|\boldsymbol{Z}^{(k)}\|^2
$$

$$
- \frac{1}{2} \lambda \eta^2 \epsilon_{k+1} \mathbb{E} \langle \boldsymbol{Z}^{(k)}, \boldsymbol{v}^{(k)} \rangle + \epsilon_{k+1} \beta^{-1} \eta d_z + (\frac{1}{2} \varsigma B^2 \eta^2 + \frac{1}{2} \varsigma B^2) \epsilon_{k+1}^2
$$

$$\leq (6M^2(\frac{1}{2}M\gamma_{k+1}^2 + \gamma_{k+1} + \frac{(3M^2 + \lambda\eta + G)\gamma_{k+1}^2}{2\delta}) + (\frac{1}{2}\varsigma M^2\eta^2 + \frac{1}{2}\varsigma M^2 + \frac{3}{2}M^2)\epsilon_{k+1}^2)\mathbb{E}\|\boldsymbol{Z}^{(k)}\|^2$$

$$+ (-(\frac{1}{2} - \frac{1}{4}\lambda)\eta\epsilon_{k+1} + (\frac{1}{2}M + \frac{1}{4}\eta^2 - \frac{1}{4}\lambda\eta^2 + 6M^2(\frac{1}{2}M\gamma_{k+1}^2 + \gamma_{k+1} + \frac{(3M^2 + \lambda\eta + G)\gamma_{k+1}^2}{2\delta}))\epsilon_{k+1}^2)\mathbb{E}\|\boldsymbol{v}^{(k)}\|^2$$

$$+ (-(\lambda\eta + G)\gamma_{k+1} + \frac{3M^2(M\delta + 3M^2 + \lambda\eta + G)\gamma_{k+1}^2}{2\delta} + (\frac{3}{2} + \frac{1}{2}\varsigma\eta^2 + \frac{1}{2}\varsigma)M^2\epsilon_{k+1}^2$$

$$+ \frac{1}{4}\eta(M^2 + 4\lambda M_0)\epsilon_{k+1})\mathbb{E}\|\boldsymbol{\theta}^{(k)} - \boldsymbol{\theta}^*\|^2 + \frac{1}{4}\eta\epsilon_{k+1}^2(\mathbb{E}\|\boldsymbol{v}^{(k)}\|^2 + 3M^2\mathbb{E}\|\boldsymbol{Z}^{(k)}\|^2 + 3M^2\mathbb{E}\|\boldsymbol{\theta}^{(k)} - \boldsymbol{\theta}^*\|^2 + 3B^2)$$

$$- \lambda\eta\epsilon_{k+1}\mathbb{E}\|F_{\boldsymbol{D}}(\boldsymbol{Z}^{(k)}, \boldsymbol{\theta}^{(k)})\| - \frac{1}{4}\lambda\eta^3\epsilon_{k+1}\mathbb{E}\|\boldsymbol{Z}^{(k)} + \eta^{-1}\boldsymbol{v}^{(k)}\|^2$$

$$+ (d_z - \frac{1}{2}A_1)\beta^{-1}\eta\epsilon_{k+1} + (\frac{1}{2}\varsigma B^2\eta^2 + \frac{1}{2}\varsigma B^2)\epsilon_{k+1}^2 + 3B^2(\frac{1}{2}M\gamma_{k+1}^2 + \gamma_{k+1} + \frac{(3M^2 + \lambda\eta + G)\gamma_{k+1}^2}{2\delta} + \frac{1}{2}\epsilon_{k+1}^2)$$

$$\leq (6M^2\gamma_{k+1}(2 + \frac{M\delta + 3M^2 + \lambda\eta + G}{\delta}\gamma_{k+1}) + (\varsigma M^2(\eta^2 + 1) + \frac{3}{2}M^2(2 + \eta))\epsilon_{k+1}^2)(\mathbb{E}\|\boldsymbol{Z}^{(k)} + \eta^{-1}\boldsymbol{v}^{(k)}\|^2 + \mathbb{E}\eta^{-2}\|\boldsymbol{v}^{(k)}\|^2)$$

$$+ (-\frac{1}{4}\eta\epsilon_{k+1} + (\frac{1}{2}M + \frac{1}{4}\eta^2 - \frac{1}{4}\lambda\eta^2 + \eta + 6M^2\gamma_{k+1}(1 + \frac{M\delta + 3M^2 + \lambda\eta + G}{2\delta})\gamma_{k+1})\epsilon_{k+1}^2)\mathbb{E}\|\boldsymbol{v}^{(k)}\|^2$$

$$+ (-(\lambda\eta + G)\gamma_{k+1} + \frac{3M^2(M\delta + 3M^2 + \lambda\eta + G)\gamma_{k+1}^2}{2\delta} + (\frac{3}{2} + \frac{1}{2}\varsigma\eta^2 + \frac{1}{2}\varsigma + \frac{3}{4}\eta)M^2\epsilon_{k+1}^2$$

$$+ \frac{1}{4}\eta(M^2 + 4\lambda M_0)\epsilon_{k+1})\mathbb{E}\|\boldsymbol{\theta}^{(k)} - \boldsymbol{\theta}^*\|^2 - \lambda\eta\epsilon_{k+1}\mathbb{E}\|F_{\boldsymbol{D}}(\boldsymbol{Z}^{(k)}, \boldsymbol{\theta}^{(k)})\| - \frac{1}{4}\lambda\eta^3\epsilon_{k+1}\mathbb{E}\|\boldsymbol{Z}^{(k)} + \eta^{-1}\boldsymbol{v}^{(k)}\|^2$$

$$+ (d_z - \frac{1}{2}A_1)\beta^{-1}\eta\epsilon_{k+1} + (\frac{1}{2}\varsigma\eta^2 + \frac{1}{2}\varsigma + \frac{3}{2} + \frac{3}{4}\eta)B^2\epsilon_{k+1}^2 + 3B^2(\frac{1}{2}M\gamma_{k+1}^2 + \gamma_{k+1} + \frac{(3M^2 + \lambda\eta + G)\gamma_{k+1}^2}{2\delta}),$$

where the first inequality is from inequalities (S19), (S18), (S17), (S16) and (S15); the second inequality uses bounds in S13 and $\mathbb{E}\|\boldsymbol{Z}^{(k+1)}\|^2 \leq 2\mathbb{E}\|\boldsymbol{Z}^{(k)}\|^2 + 2\epsilon_{k+1}^2\mathbb{E}\|\boldsymbol{v}^{(k)}\|^2$; the third inequality uses $2\mathbb{E}\langle\boldsymbol{v}^{(k)}, \nabla_{\boldsymbol{Z}}F_{\boldsymbol{D}}(\boldsymbol{Z}^{(k)}, \boldsymbol{\theta}^{(k)})\rangle \leq \mathbb{E}\|\boldsymbol{v}^{(k)}\|^2 + \mathbb{E}\|\nabla_{\boldsymbol{Z}}F_{\boldsymbol{D}}(\boldsymbol{Z}^{(k)}, \boldsymbol{\theta}^{(k)})\|^2$, the bound in (S13) and the dissipative condition in (S14); and the last inequality uses $\mathbb{E}\|\boldsymbol{Z}^{(k)}\|^2 \leq 2\mathbb{E}\|\boldsymbol{Z}^{(k)} + \eta^{-1}\boldsymbol{v}^{(k)}\|^2 + 2\mathbb{E}\eta^{-2}\|\boldsymbol{v}^{(k)}\|^2$.

For notational simplicity, we can define

$$G_0 = \frac{\delta\lambda\eta}{3M^2 + \lambda\eta + G},$$

$$G_1 = \frac{1}{4}\eta,$$

$$G_2 = \eta^{-2}(\varsigma M^2(\eta^2 + 1) + \frac{3}{2}M^2(2 + \eta)) + \frac{1}{2}M + \frac{1}{4}\eta^2 - \frac{1}{4}\lambda\eta^2 + \eta + 6M^2(1 + \frac{M\delta + 3M^2 + \lambda\eta + 1}{2\delta}),$$

$$G_3 = 6M^2\eta^{-2}(2 + \frac{M\delta + 3M^2 + \lambda\eta + 1}{\delta}) + \frac{\delta\lambda\eta}{4(3M^2 + \lambda\eta)},$$

$$G_4 = \frac{1}{2}\lambda\eta,$$

$$G_5 = \frac{3M^2(M\delta + 3M^2 + \lambda\eta + 1)}{2\delta},$$

$$G_6 = \frac{3M^2}{2} + \frac{1}{2}\varsigma M^2\eta^2 + \frac{1}{2}\varsigma M^2 + \frac{3}{4}M^2\eta,$$

$$G_7 = \frac{1}{4}\eta(M^2 + 4\lambda M_0)$$

$$G_8 = \lambda\eta,$$

$$G_9 = \frac{1}{4}\lambda\eta^3,$$

$$G_{10} = \frac{\delta\lambda\eta}{3M^2 + \lambda\eta},$$

$$G_{11} = \varsigma M^2(\eta^2 + 1) + \frac{3}{2}M^2(2 + \eta),$$

$$G_{12} = 6M^2\eta^{-2}(2 + \frac{M\delta + 3M^2 + \lambda\eta + 1}{\delta}) + \frac{\delta\lambda\eta^3}{4(3M^2 + \lambda\eta)},$$

$$G_{13} = (d_z - \frac{1}{2}A_1)\beta^{-1}\eta + (\frac{1}{2}\varsigma\eta^2 + \frac{1}{2}\varsigma + \frac{3}{2} + 3\eta)B^2 + 3B^2(\frac{1}{2}M + 1 + \frac{(3M^2 + \lambda\eta + 1)}{2\delta}).$$

Consider decaying step size sequences $\epsilon_k = \frac{C_\epsilon}{c_e + k^\alpha}, \gamma_k = \frac{C_\gamma}{c_g + k^\alpha}$ for some constants $\alpha \in (0,1)$, $c_e \geq 0$ and $c_g \geq 0$, where

$$C_\epsilon = \min\left\{1, \frac{G_1}{2G_2}, \frac{G_{10}}{2G_{11}}\right\}, \quad C_\gamma = \min\left\{1, \frac{G_1 C_\epsilon}{2G_3}, \frac{G_1 C_\epsilon c_g}{2G_3 c_e}, \frac{G_8 C_\epsilon}{G_9}, \frac{G_8 C_\epsilon c_g}{G_9 c_e}, \frac{G_{10} C_\epsilon}{2G_{12}}, \frac{G_{10} C_\epsilon c_g}{2G_{12} c_e}, \left(\frac{G_4}{2G_5}\right)^2\right\}.$$

Let $G = \max\{\frac{G_7 C_\epsilon}{C_\gamma}, \frac{G_7 C_\epsilon c_g}{C_\gamma} c_e\}$, and let $k_0$ be an integer such that $c_e + (k_0 + 1)^\alpha > \max\{\frac{2G_6 C_\epsilon^2}{G_4 C_\gamma}, \frac{2G_6 C_\epsilon^2 c_g}{G_4 C_\gamma c_e}\}$ and $c_g + (k_0 + 1)^\alpha > GC_\gamma^2$. Then for $k \geq k_0$, we have

$$L(k+1) - L(k) \leq -G_0 \gamma_{k+1} L(k) + (-G_1 \epsilon_{k+1} + G_2 \epsilon_{k+1}^2 + G_3 \gamma_{k+1})\mathbb{E}\|\boldsymbol{v}^{(k)}\|^2$$
$$+ (-(G_4 + G)\gamma_{k+1} + G_5 \gamma_{k+1}^{\frac{3}{2}} + G_6 \epsilon_{k+1}^2 + G_7 \epsilon_{k+1})\mathbb{E}\|\boldsymbol{\theta}^{(k)} - \boldsymbol{\theta}^*\|^2 + (-G_8 \epsilon_{k+1} + G_9 \gamma_{k+1})\mathbb{E}\|F_{\boldsymbol{D}}(\boldsymbol{Z}^{(k)}, \boldsymbol{\theta}^{(k)})\|$$
$$+ (-G_{10} \epsilon_{k+1} + G_{11} \epsilon_{k+1}^2 + G_{12} \gamma_{k+1})\mathbb{E}\|\boldsymbol{Z}^{(k)} + \eta^{-1}\boldsymbol{v}^{(k)}\|^2 + G_{13} \epsilon_{k+1},$$

and

$$- G_1 \epsilon_{k+1} + G_2 \epsilon_{k+1}^2 + G_3 \gamma_{k+1} \leq 0,$$
$$- (G_4 + G)\gamma_{k+1} + G_5 \gamma_{k+1}^{\frac{3}{2}} + G_6 \epsilon_{k+1}^2 + G_7 \epsilon_{k+1} \leq 0,$$
$$- G_8 \epsilon_{k+1} + G_9 \gamma_{k+1} \leq 0,$$
$$- G_{10} \epsilon_{k+1} + G_{11} \epsilon_{k+1}^2 + G_{12} \gamma_{k+1} \leq 0.$$

Let $C_L = \max\{\frac{G_{13} C_\epsilon}{G_0 C_\gamma}, \frac{G_{13} C_\epsilon c_g}{G_0 C_\gamma c_e}, L(0), L(1), \ldots, L(k_0)\}$, we can prove by induction that $L(k) \leq C_L$ for all $k$.

By the definition of $C_L$, $L(k) \leq C_L$ for all $k \leq k_0$. Assume that $L(i) \leq C_L$ for all $i \leq k$ for some $k \geq k_0$. Then we have

$$L(k+1) \leq L(k) - G_0 \gamma_{k+1} L(k) + G_{13} \epsilon_{k+1} \leq C_L - G_0 \frac{G_{13} C_\epsilon}{G_0 C_\gamma} \gamma_{k+1} + G_{13} \epsilon_{k+1} \leq C_L.$$

By induction, we have $L(k) \leq C_L$ for all $k$.

Then, by inequality (S20), we can give uniform $L_2$ bounds for $\mathbb{E}\|\boldsymbol{\theta}^{(k)}\|^2, \mathbb{E}\|\boldsymbol{v}^{(k)}\|^2$ and $\mathbb{E}\|\boldsymbol{Z}^{(k)}\|^2$: there exist constants $C_{\boldsymbol{\theta}} = \frac{2\delta C_L}{3M^2 + \lambda\eta + G}$, $C_{\boldsymbol{Z}} = \frac{8C_L}{(1-2\lambda)\eta^2}$, $C_{\boldsymbol{v}} = \frac{4C_L}{1-2\lambda}$ such that $\sup_{i \geq 0} \mathbb{E}\left\|\boldsymbol{v}^{(i)}\right\|^2 \leq C_{\boldsymbol{v}}$, $\sup_{i \geq 0} \mathbb{E}\left\|\boldsymbol{Z}^{(i)}\right\|^2 \leq C_{\boldsymbol{Z}}$, and $\sup_{i \geq 0} \mathbb{E}\left\|\boldsymbol{\theta}^{(i)}\right\|^2 \leq C_{\boldsymbol{\theta}}$ hold. The proof is completed. $\square$

**Remark S1** *As pointed out in the proof of Theorem S1, the values of $\lambda_0'$ and $k_0$ depend only on $\delta$ and the sequence $\{\gamma_k\}$. The second term of $\lambda_0$ characterizes the effects of the constants $(M, B, m, b, \delta, \zeta_2)$ defined in the assumptions, the friction coefficient $\eta$, the learning rate sequence $\{\epsilon_k\}$, and the step size sequence $\{\gamma_k\}$ on the convergence of $\boldsymbol{\theta}^{(k)}$. In particular, $\eta$, $\{\epsilon_k\}$, and $\{\gamma_k\}$ affects on the convergence of $\boldsymbol{\theta}^{(k)}$ via the upper bounds $C_{\boldsymbol{\theta}}$ and $C_{\boldsymbol{Z}}$.*

### 3.2 Proof of Theorem 3.2

The convergence of $\boldsymbol{Z}^{(k)}$ is studied in terms of the *2-Wasserstein distance* defined by

$$\mathcal{W}_2(\mu, \nu) := \inf\{(\mathbb{E}\|\boldsymbol{Z} - \boldsymbol{Z}'\|^2)^{1/2} : \mu = \mathcal{L}(\boldsymbol{Z}), \nu = \mathcal{L}(\boldsymbol{Z}')\},$$

where $\mu$ and $\nu$ are Borel probability measures on $\mathbb{R}^{d_z}$ with finite second moments, and the infimum is taken over all random couples $(\boldsymbol{Z}, \boldsymbol{Z}')$ taking values from $\mathbb{R}^{d_z} \times \mathbb{R}^{d_z}$ with marginals $\boldsymbol{Z} \sim \mu$ and $\boldsymbol{Z}' \sim \nu$. To complete the proof, we make the following assumption for the initial distribution of $\boldsymbol{Z}^{(0)}$:

**Assumption B7** *The probability law $\mu_0$ of the initial value $\boldsymbol{Z}^{(0)}$ has a bounded and strictly positive density $p_0$ with respect to the Lebesgue measure, and $\kappa_0 := \log \int e^{\|\boldsymbol{Z}\|^2} p_0(\boldsymbol{Z})d\boldsymbol{Z} < \infty$.*

Recall that for the purpose of sufficient dimension reduction, we need to consider the convergence of Algorithm 1 under the case that the full dataset is used at each iteration. In this case, the discrete-time Markov process (16) can be viewed as a discretization of the continuous-time underdamped Langevin diffusion at a fixed value of $\boldsymbol{\theta}$, i.e.,

$$
\begin{aligned}
d\boldsymbol{v}(t) &= -\eta\boldsymbol{v}(t)dt - \nabla_{\boldsymbol{Z}}F_{\boldsymbol{D}}(\boldsymbol{Z}(t),\boldsymbol{\theta})dt + \sqrt{2\eta/\beta}dB(t), \\
d\boldsymbol{Z}(t) &= \boldsymbol{v}(t)dt,
\end{aligned}
\tag{S21}
$$

where $\{B(t)\}_{t\geq 0}$ is the standard Brownian motion in $\mathbb{R}^{d_z}$.

Let $\mu_{\boldsymbol{D},k}$ denote the probability law of $(\boldsymbol{Z}^{(k)},\boldsymbol{v}^{(k)})$ given the dataset $\boldsymbol{D}$, let $\nu_{\boldsymbol{D},t}$ denote the probability law of $(\boldsymbol{Z}(t),\boldsymbol{v}(t))$ following the process above, and let $\pi_{\boldsymbol{D}}$ denote the stationary distribution of the process. Following Gao et al. (2021), we will first show that the SGHMC sample $(\boldsymbol{Z}^{(k)},\boldsymbol{v}^{(k)})$ tracks the continuous time underdamped Langevin diffusion in *2-Wasserstein distance*. With the convergence of the diffusion to $\pi_{\boldsymbol{D}}$, we will then be able to estimate the *2-Wasserstein distance* $\mathcal{W}(\mu_{\boldsymbol{D},k},\pi_{\boldsymbol{D}})$.

Let $T_k = \sum_{i=0}^{k-1}\epsilon_{i+1}$. Following the proof of Lemma 18 in Gao et al. (2021), we have Theorem S2, which provides an upper bound for $\mathcal{W}_2(\mu_{\boldsymbol{D},k},\nu_{\boldsymbol{D},T_k})$.

**Theorem S2** *Suppose Assumptions B1-B7 hold. Then for any $k \in \mathbb{N}$,*

$$
\mathcal{W}_2(\nu_{\boldsymbol{D},T_k},\mu_{\boldsymbol{D},T_k}) \leq \sqrt{C_5\log(T_k)}\left(\sqrt{\tilde{C}(k)} + \left(\frac{\tilde{C}(k)}{2}\right)^{1/4}\right) + \sqrt{C_6T_k\sum_{j=1}^{k-1}\epsilon_{j+1}^2},
$$

$$
\text{where} \quad \tilde{C}(k) = C_1T_k^2\sum_{j=1}^{k-1}\epsilon_{j+1}^2 + C_2\sum_{j=1}^{k-1}\epsilon_{j+1}\gamma_j + C_3\varsigma T_k + C_4\sum_{j=1}^{k-1}\epsilon_{j+1}^2,
\tag{S22}
$$

*and $C_1$, $C_2$, $C_3$, $C_4$, $C_5$, $C_6$ are some constants.*

PROOF: Our proof follows the proof of Lemma 18 in Gao et al. (2021). Recall that $T_k = \sum_{i=1}^{k}\epsilon_k$. Let $\bar{T}(s) = T_k$ for $T_k \leq s < T_{k+1}, k = 1,\ldots,\infty$. We first consider an auxiliary diffusion process $(\tilde{Z}(t),\tilde{V}(t))$:

$$
\begin{aligned}
\tilde{\boldsymbol{v}}(t) =&\, \boldsymbol{v}(0) - \int_0^t \eta\tilde{\boldsymbol{v}}(\bar{T}(s))ds \\
&- \int_0^t \nabla_{\boldsymbol{Z}}\hat{F}_{\boldsymbol{D}}\left(\boldsymbol{Z}(0) + \int_0^{\bar{T}(s)}\tilde{\boldsymbol{v}}(\bar{T}(u))du, \bar{\boldsymbol{\theta}}(s)\right)ds + \sqrt{2\eta\beta^{-1}}\int_0^t dB(s),
\end{aligned}
\tag{S23}
$$

$$
\tilde{\boldsymbol{Z}}(t) =\, \boldsymbol{Z}(0) + \int_0^t \tilde{\boldsymbol{v}}(s)ds,
\tag{S24}
$$

where $\bar{\boldsymbol{\theta}}(s) = \boldsymbol{\theta}_k$ for $T_k \leq s < T_{k+1}$. By the definition of $\tilde{\boldsymbol{v}}(t)$, $\left(\boldsymbol{Z}(0) + \int_0^{T_k}\tilde{\boldsymbol{v}}(\bar{T}(s))ds, \tilde{\boldsymbol{v}}(T_k)\right)$ has the same law as $\mu_{\boldsymbol{D},k}$. Let $\mathbb{P}$ be the probability measure associated with the underdamped Langevin diffusion $(\boldsymbol{Z}(t),\boldsymbol{v}(t))$ and $\tilde{\mathbb{P}}$ be the probability measure associated with the $(\tilde{\boldsymbol{Z}}(t),\tilde{\boldsymbol{v}}(t))$ process. Let $\mathcal{F}_t$ denote the natural filtration up to time $t$. Then by the Girsanov theorem, the Radon-Nikodym derivative of $\mathbb{P}$ w.r.t. $\tilde{\mathbb{P}}$ is given by

$$
\left.\frac{d\mathbb{P}}{d\tilde{\mathbb{P}}}\right|_{\mathcal{F}_t} = e^{-\sqrt{\frac{\beta}{2\eta}}\int_0^t\left(\eta\tilde{\boldsymbol{v}}(s) - \eta\tilde{\boldsymbol{v}}(\bar{T}(s)) + \nabla_{\boldsymbol{Z}}F_{\boldsymbol{D}}(\tilde{Z}(s),\boldsymbol{\theta}^*) - \nabla_{\boldsymbol{Z}}\hat{F}_{\boldsymbol{D}}\left(\boldsymbol{Z}(0)+\int_0^{\bar{T}(s)}\tilde{\boldsymbol{v}}(\bar{T}(u))du,\bar{\boldsymbol{\theta}}(s)\right)\right)\cdot dB(s)}
$$

$$
\cdot\, e^{-\frac{\beta}{4\eta}\int_0^t\left\|\eta\tilde{\boldsymbol{v}}(s) - \eta\tilde{\boldsymbol{v}}(\bar{T}(s)) + \nabla_{\boldsymbol{Z}}F_{\boldsymbol{D}}(\tilde{\boldsymbol{Z}}(s),\boldsymbol{\theta}^*) - \nabla_{\boldsymbol{Z}}\hat{F}_{\boldsymbol{D}}\left(\boldsymbol{Z}(0)+\int_0^{\bar{T}(s)}\tilde{\boldsymbol{v}}(\bar{T}(u))du,\bar{\boldsymbol{\theta}}(s)\right)\right\|^2 ds}.
$$

Let $\mathbb{P}_t$ and $\tilde{\mathbb{P}}_t$ denote the probability measures $\mathbb{P}$ and $\tilde{\mathbb{P}}$ conditional on the filtration $\mathcal{F}_t$. Then

$$D(\tilde{\mathbb{P}}_t \| \mathbb{P}_t) := -\int d\tilde{\mathbb{P}}_t \log \frac{d\mathbb{P}_t}{d\tilde{\mathbb{P}}_t}$$

$$= \frac{\beta}{4\eta} \int_0^t \mathbb{E} \left\| \eta \tilde{\boldsymbol{v}}(s) - \eta \tilde{\boldsymbol{v}}(\bar{T}(s)) + \nabla_{\boldsymbol{Z}} F_{\boldsymbol{D}}(\tilde{\boldsymbol{Z}}(s), \boldsymbol{\theta}^*) - \nabla_{\boldsymbol{Z}} \hat{F}_{\boldsymbol{D}} \left( Z(0) + \int_0^{\bar{T}(s)} \tilde{\boldsymbol{v}}(\bar{T}(u)) du, \bar{\boldsymbol{\theta}}(s) \right) \right\|^2 ds$$

$$\leq \frac{\beta}{2\eta} \int_0^t \mathbb{E} \left\| \nabla_{\boldsymbol{Z}} F_{\boldsymbol{D}} \left( \boldsymbol{Z}(0) + \int_0^{\bar{T}(s)} \tilde{\boldsymbol{v}}(u) du, , \boldsymbol{\theta}^* \right) - \nabla_{\boldsymbol{Z}} \hat{F}_{\boldsymbol{D}} \left( \boldsymbol{Z}(0) + \int_0^{\bar{T}(s)} \tilde{\boldsymbol{v}}(\bar{T}(u)) du, \bar{\boldsymbol{\theta}}(s) \right) \right\|^2 ds$$

$$+ \frac{\beta}{2\eta} \int_0^t \mathbb{E} \left\| \eta \tilde{\boldsymbol{v}}(s) - \eta \tilde{\boldsymbol{v}}(\bar{T}(s)) \right\|^2 ds$$

$$\leq \frac{3\beta}{2\eta} \int_0^t \mathbb{E} \left\| \nabla_{\boldsymbol{Z}} F_{\boldsymbol{D}} \left( \boldsymbol{Z}(0) + \int_0^{\bar{T}(s)} \tilde{\boldsymbol{v}}(u) du, \boldsymbol{\theta}^* \right) - \nabla_{\boldsymbol{Z}} F_{\boldsymbol{D}} \left( \boldsymbol{Z}(0) + \int_0^{\bar{T}(s)} \tilde{\boldsymbol{v}}(\bar{T}(u)) du, \boldsymbol{\theta}^* \right) \right\|^2 ds$$

$$+ \frac{3\beta}{2\eta} \int_0^t \mathbb{E} \left\| \nabla_{\boldsymbol{Z}} F_{\boldsymbol{D}} \left( \boldsymbol{Z}(0) + \int_0^{\bar{T}(s)} \tilde{\boldsymbol{v}}(\bar{T}(u)) du, , \boldsymbol{\theta}^* \right) - \nabla_{\boldsymbol{Z}} F_{\boldsymbol{D}} \left( \boldsymbol{Z}(0) + \int_0^{\bar{T}(s)} \tilde{\boldsymbol{v}}(\bar{T}(u)) du, \bar{\boldsymbol{\theta}}(s) \right) \right\|^2 ds$$

$$+ \frac{3\beta}{2\eta} \int_0^t \mathbb{E} \left\| \nabla_{\boldsymbol{Z}} F_{\boldsymbol{D}} \left( \boldsymbol{Z}(0) + \int_0^{\bar{T}(s)} \tilde{\boldsymbol{v}}(\bar{T}(u)) du, \bar{\boldsymbol{\theta}}(s) \right) - \nabla_{\boldsymbol{Z}} \hat{F}_{\boldsymbol{D}} \left( \boldsymbol{Z}(0) + \int_0^{\bar{T}(s)} \tilde{\boldsymbol{v}}(\bar{T}(u)) du, \bar{\boldsymbol{\theta}}(s) \right) \right\|^2 ds$$

$$+ \frac{\beta}{2\eta} \int_0^t \mathbb{E} \left\| \eta \tilde{\boldsymbol{v}}(s) - \eta \tilde{\boldsymbol{v}}(\bar{T}(s)) \right\|^2 ds,$$

which implies

$$D(\tilde{\mathbb{P}}_{T_k} \| \mathbb{P}_{T_k})$$

$$\leq \frac{3\beta}{2\eta} \sum_{j=0}^{k-1} \epsilon_{j+1} \mathbb{E}_{\boldsymbol{D}} \left\| \nabla_{\boldsymbol{Z}} F_{\boldsymbol{D}} \left( \boldsymbol{Z}(0) + \int_0^{T_j} \tilde{\boldsymbol{v}}(u) du, \boldsymbol{\theta}^* \right) - \nabla_{\boldsymbol{Z}} F_{\boldsymbol{D}} \left( \boldsymbol{Z}(0) + \int_0^{T_j} \tilde{\boldsymbol{v}}(\bar{T}(u)) du, \boldsymbol{\theta}^* \right) \right\|^2$$

$$+ \frac{3\beta}{2\eta} \sum_{j=0}^{k-1} \epsilon_{j+1} \mathbb{E} \left\| \nabla_{\boldsymbol{Z}} F_{\boldsymbol{D}} \left( \boldsymbol{Z}(0) + \int_0^{T_j} \tilde{\boldsymbol{v}}(\bar{T}(u)) du, \boldsymbol{\theta}^* \right) - \nabla_{\boldsymbol{Z}} F_{\boldsymbol{D}} \left( \boldsymbol{Z}(0) + \int_0^{T_j} \tilde{\boldsymbol{v}}(\bar{T}(u)) du, \boldsymbol{\theta}^{(j)} \right) \right\|^2$$

$$+ \frac{3\beta}{2\eta} \sum_{j=0}^{k-1} \epsilon_{j+1} \mathbb{E} \left\| \nabla_{\boldsymbol{Z}} \hat{F}_{\boldsymbol{D}} \left( \boldsymbol{Z}(0) + \int_0^{T_j} \tilde{\boldsymbol{v}}(\bar{T}(u)) du, \boldsymbol{\theta}^{(j)} \right) - \nabla_{\boldsymbol{Z}} F_{\boldsymbol{D}} \left( \boldsymbol{Z}(0) + \int_0^{T_j} \tilde{\boldsymbol{v}}(\bar{T}(u)) du, \boldsymbol{\theta}^{(j)} \right) \right\|^2$$

$$+ \frac{\beta}{2\eta} \sum_{j=0}^{k-1} \int_{T_j}^{T_{j+1}} \mathbb{E} \left\| \eta \tilde{\boldsymbol{v}}(s) - \eta \tilde{\boldsymbol{v}}(\bar{T}(s)) \right\|^2 ds$$

$$= (I) + (II) + (III) + (IV).$$

$$\text{(S25)}$$

We first bound the term (I) in (S25):

$$(I) \leq \frac{3\beta}{2\eta} \sum_{j=0}^{k-1} M^2 \epsilon_{j+1} \mathbb{E} \left\| \int_0^{T_j} \left( \tilde{\boldsymbol{v}}(u) - \tilde{\boldsymbol{v}}(\bar{T}(u)) \right) du \right\|^2 \leq \frac{3\beta}{2\eta} \sum_{j=0}^{k-1} M^2 \epsilon_{j+1} T_j \int_0^{T_j} \mathbb{E} \left\| \tilde{\boldsymbol{v}}(u) - \tilde{\boldsymbol{v}}(\bar{T}(u)) \right\|^2 du$$

$$= \frac{3\beta}{2\eta} \sum_{j=0}^{k-1} M^2 \epsilon_{j+1} T_j \sum_{i=0}^{j-1} \int_{T_i}^{T_{i+1}} \mathbb{E} \left\| \tilde{\boldsymbol{v}}(u) - \tilde{\boldsymbol{v}}(\bar{T}(u)) \right\|^2 du.$$

For $T_i < u \leq T_{i+1}$, we have

$$\tilde{\boldsymbol{v}}(u) - \tilde{\boldsymbol{v}}(\bar{T}(u)) = -(u - T_i) \eta \boldsymbol{v}^{(i)} - (u - T_i) \nabla_{\boldsymbol{Z}} F_{\boldsymbol{D}} \left( \boldsymbol{Z}^{(i)}, \boldsymbol{\theta}^{(i)} \right) + \sqrt{2\eta\beta^{-1}} (B(u) - B(T_i)), \quad \text{(S26)}$$

in distribution. Therefore,

$$
\mathbb{E}\left\|\tilde{\boldsymbol{v}}(u) - \tilde{\boldsymbol{v}}(\bar{T}(u))\right\|^2
$$

$$
= (u - T_i)^2 \mathbb{E}\left\|\eta \boldsymbol{v}^{(i)} + \nabla_{\boldsymbol{Z}} F_{\boldsymbol{D}}\left(\boldsymbol{Z}^{(i)}, \boldsymbol{\theta}^{(i)}\right)\right\|^2 + 2\eta\beta^{-1}(u - T_i)
$$

$$
= (u - T_i)^2 \mathbb{E}\left\|\eta \boldsymbol{v}^{(i)} + \nabla_{\boldsymbol{Z}} F_{\boldsymbol{D}}\left(\boldsymbol{Z}^{(i)}, \boldsymbol{\theta}^*\right)\right\|^2 + (u - T_i)^2 \mathbb{E}\left\|\nabla_{\boldsymbol{Z}} F_{\boldsymbol{D}}\left(\boldsymbol{Z}^{(i)}, \boldsymbol{\theta}^{(i)}\right) - \nabla_{\boldsymbol{Z}} F_{\boldsymbol{D}}\left(\boldsymbol{Z}^{(i)}, \boldsymbol{\theta}^*\right)\right\|^2
$$

$$
\quad + 2\eta\beta^{-1}(u - T_i)
$$

$$
\leq 2\epsilon_{i+1}^2 \mathbb{E}\left\|\eta \boldsymbol{v}^{(i)}\right\|^2 + 2\epsilon_{i+1}^2 \mathbb{E}\left\|\nabla_{\boldsymbol{Z}} F_{\boldsymbol{D}}\left(\boldsymbol{Z}^{(i)}, \boldsymbol{\theta}^*\right)\right\|^2 + \epsilon_{i+1}^2(M^2 \mathbb{E}\left\|\boldsymbol{\theta}^{(i)} - \boldsymbol{\theta}^*\right\|^2) + 2\eta\beta^{-1}\epsilon_{i+1}
$$

$$
\leq 2\eta^2 \epsilon_{i+1}^2 \mathbb{E}\left\|\boldsymbol{v}^{(i)}\right\|^2 + 6\epsilon_{i+1}^2 \left(M^2 \mathbb{E}\left\|\boldsymbol{Z}^{(i)}\right\|^2 + B^2\right) + \lambda_0 M^2 \epsilon_{i+1}^2 \gamma_i + 2\eta\beta^{-1}\epsilon_{i+1}. \qquad (S27)
$$

This implies

$$
(I) \leq \frac{3\beta}{2\eta} \sum_{j=0}^{k-1} M^2 \epsilon_{j+1} T_j \sum_{i=0}^{j-1} \int_{T_i}^{T_{i+1}} \mathbb{E}\left\|\tilde{\boldsymbol{v}}(u) - \tilde{\boldsymbol{v}}(\bar{T}(u))\right\|^2 du
$$

$$
\leq \frac{3M^2\beta}{2\eta} \sum_{j=0}^{k-1} \epsilon_{j+1} T_j \sum_{i=0}^{j-1} \left(2\eta^2 \epsilon_{i+1}^3 \sup_{i\geq 0}\mathbb{E}\left\|\boldsymbol{v}^{(i)}\right\|^2 + 6\epsilon_{i+1}^3 \left(M^2 \sup_{i\geq 0}\mathbb{E}\left\|\boldsymbol{Z}^{(i)}\right\|^2 + B^2\right) + \lambda_0 M^2 \epsilon_{i+1}^3 \gamma_i + 2\eta\beta^{-1}\epsilon_{i+1}^2\right).
$$

We can bound the term (II) in (S25):

$$
(II) \leq \frac{3\beta}{2\eta} \sum_{j=0}^{k-1} \epsilon_{j+1} M^2 \mathbb{E}\left\|\boldsymbol{\theta}^{(j)} - \boldsymbol{\theta}^*\right\|^2 \leq \frac{3\lambda_0 M^2 \beta}{2\eta} \sum_{j=0}^{k-1} \epsilon_{j+1}\gamma_j.
$$

We can bound the term (III) in (S25):

$$
(III) \leq \frac{3\beta}{2\eta} \sum_{j=0}^{k-1} \epsilon_{j+1} 2\varsigma \left(M^2 \left\|\boldsymbol{Z}(0) + \int_0^{T_j} \tilde{\boldsymbol{v}}(\bar{T}(u))du, \boldsymbol{\theta}^{(j)}\right\|^2 + M^2 \mathbb{E}\left\|\boldsymbol{\theta}^{(j)} - \boldsymbol{\theta}^*\right\|^2 + B^2\right)
$$

$$
= \frac{3\beta}{2\eta} \sum_{j=0}^{k-1} \epsilon_{j+1} 2\varsigma \left(M^2 \mathbb{E}\left\|\boldsymbol{Z}^{(j)}\right\|^2 + M^2 \mathbb{E}\left\|\boldsymbol{\theta}^{(j)} - \boldsymbol{\theta}^*\right\|^2 + B^2\right)
$$

$$
\leq \frac{3\beta}{2\eta} \sum_{j=0}^{k-1} \epsilon_{j+1} 2\varsigma \left(M^2 \sup_{i\geq 0}\mathbb{E}\left\|\boldsymbol{Z}^{(i)}\right\|^2 + M^2 \sup_{i\geq 0}\mathbb{E}\left\|\boldsymbol{\theta}^{(j)} - \boldsymbol{\theta}^*\right\|^2 + B^2\right).
$$

Finally, let us bound the term (IV) in (S25) as follows:

$$
(IV) \leq \frac{\beta\eta}{2} \sum_{j=0}^{k-1} \left(2\eta^2 \epsilon_{j+1}^3 \sup_{\geq 0}\mathbb{E}\left\|\boldsymbol{v}^{(i)}\right\|^2 + 6\epsilon_{j+1}^3 \left(M^2 \sup_{i\geq 0}\mathbb{E}\left\|\boldsymbol{Z}^{(i)}\right\|^2 + B^2\right) + \lambda_0 M^2 \epsilon_{j+1}^3 \gamma_i + 2\eta\beta^{-1}\epsilon_{j+1}^2\right),
$$

where the estimate in (S27) is used.

In the proof of Theorem 3.1, we have shown that $\mathbb{E}\left\|\boldsymbol{v}^{(j)}\right\|^2$, $\mathbb{E}\left\|\boldsymbol{Z}^{(j)}\right\|^2$ and $\mathbb{E}\left\|\boldsymbol{\theta}^{(j)} - \boldsymbol{\theta}^*\right\|^2$ are bounded by some constants $C_{\boldsymbol{v}}$, $C_{\boldsymbol{Z}}$ and $C_{\boldsymbol{\theta}}$. Then for decaying step size sequence $\{\epsilon_{k+1}\}$ and $\{\gamma_{k+1}\}$ with $\epsilon_0 < 1$ and $\gamma_0 < 1$, there exists some constant $C_1, C_2, C_3$ such that

$$
D(\tilde{\mathbb{P}}_{T_k} \| \mathbb{P}_{T_k}) \leq C_1 T_k^2 \sum_{j=1}^{k-1} \epsilon_{j+1}^2 + C_2 \sum_{j=1}^{k-1} \epsilon_{j+1}\gamma_j + C_3 \varsigma T_k + C_4 \sum_{j=1}^{k-1} \epsilon_{j+1}^2 := \tilde{C}(k),
$$

where

$$C_1 = \frac{3M^2\beta}{2\eta}(2\eta^2 C_{\boldsymbol{v}} + 6M^2 C_{\boldsymbol{Z}} + 6B^2 + 2\eta\beta^{-1}),$$

$$C_2 = \frac{3\lambda_0 M^2 \beta}{2\eta},$$

$$C_3 = \frac{3\beta}{\eta}(M^2 C_{\boldsymbol{Z}} + M^2 C_{\boldsymbol{\theta}} + B^2),$$

$$C_4 = \frac{\beta\eta}{2}(2\eta^2 C_{\boldsymbol{v}} + 6M^2 C_{\boldsymbol{Z}} + 6B^2 + \lambda_0 M^2 + 2\eta\beta^{-1}).$$

(S28)

For any two Borel probability measures $\mu, \nu$ on $\mathbb{R}^{2d}$ with finite second moments, we can apply the result of Bolley and Villani (2005) to connect $\mathcal{W}_2(\mu, \nu)$ and $D(\mu\|\nu)$:

$$\mathcal{W}_2(\mu, \nu) \leq C_\nu \left[ \sqrt{D(\mu\|\nu)} + \left( \frac{D(\mu\|\nu)}{2} \right)^{1/4} \right],$$

where

$$C_\nu = 2 \inf_{\lambda > 0} \left( \frac{1}{\lambda} \left( \frac{3}{2} + \log \int_{\mathbb{R}^{2d}} e^{\lambda\|w\|^2} \nu(dw) \right) \right)^{1/2}.$$

Using the results in Lemma 17 and Lemma 18 of Gao et al. (2021), we have $C^2_{\nu_{\boldsymbol{D}, T_k}} \leq C_5 \log(T_k)$ for some constant

$$C_5 = \frac{2\sqrt{2}}{\sqrt{\alpha_0}} \left( \frac{5}{2} + \log \left( \int_{\mathbb{R}^{2d_z}} e^{\frac{1}{4}\alpha \mathcal{V}(\boldsymbol{Z}, \boldsymbol{v})} \mu_0(d\boldsymbol{Z}, d\boldsymbol{v}) + \frac{1}{4} e^{\frac{\alpha(d_z + A_1)}{3\lambda}} \alpha\eta(d_z + A_1) \right) \right)^{1/2},$$

where $\alpha = \frac{\lambda(1-2\lambda)}{12}$, $\alpha_0 = \frac{\alpha}{\frac{64}{(1-2\lambda)\beta\eta^2} + \frac{32}{\beta(1-2\lambda)}}$, and the Lyapunov function

$$\mathcal{V}(\boldsymbol{Z}, \boldsymbol{v}) := \beta F_{\boldsymbol{D}}(\boldsymbol{Z}, \boldsymbol{\theta}^*) + \frac{\beta}{4}\eta^2(\|\boldsymbol{Z} + \eta^{-1}\boldsymbol{v}\|^2 + \|\eta^{-1}\boldsymbol{v}\|^2 - \lambda\|\boldsymbol{Z}\|^2). \tag{S29}$$

Then we have

$$\mathcal{W}_2(\tilde{\mathbb{P}}_{T_k}, \nu_{\boldsymbol{D}, T_k}) \leq \sqrt{C_5 \log(T_k)} \left( \sqrt{\tilde{C}(k)} + \left( \frac{\tilde{C}(k)}{2} \right)^{1/4} \right).$$

Finally, let us provide a bound for $\mathcal{W}_2(\mu_{\boldsymbol{D}, k}, \tilde{\mathbb{P}}_{T_k})$. Note that by the definition of $\tilde{V}$, we have that $\left( Z(0) + \int_0^{T_k} \tilde{\boldsymbol{v}}(\bar{T}(s))ds, \tilde{\boldsymbol{v}}(T_k) \right)$ has the same law as $\mu_{\boldsymbol{z}, k}$, and we can compute that

$$\mathbb{E} \left\| \tilde{\boldsymbol{Z}}(T_k) - \boldsymbol{Z}(0) - \int_0^{T_k} \tilde{\boldsymbol{v}}(\bar{T}(s))ds \right\|^2 = \mathbb{E} \left\| \int_0^{T_k} \tilde{\boldsymbol{v}}(s) - \tilde{\boldsymbol{v}}(\bar{T}(s))ds \right\|^2$$

$$\leq T_k \int_0^{T_k} \mathbb{E} \left\| \tilde{\boldsymbol{v}}(s) - \tilde{\boldsymbol{v}}(\bar{T}(s)) \right\|^2 ds$$

$$\leq T_k \sum_{k=0}^{j-1} \left( 2\eta^2 \epsilon_{i+1}^3 \sup_{i \geq 0} \mathbb{E} \left\| \boldsymbol{v}^{(i)} \right\|^2 + 6\epsilon_{i+1}^3 \left( M^2 \sup_{i \geq 0} \mathbb{E} \left\| \boldsymbol{Z}^{(i)} \right\|^2 + B^2 \right) + \lambda_0 M^2 \epsilon_{i+1}^3 \gamma_i + 2\eta\beta^{-1}\epsilon_{i+1}^2 \right)$$

$$\leq C_6 T_k \sum_{j=1}^{k-1} \epsilon_{j+1}^2,$$

where constant $C_6 = 2\eta^2 C_{\boldsymbol{v}} + 6M^2 C_{\boldsymbol{Z}} + 6B^2 + \lambda_0 M^2 + 2\eta\beta^{-1}$. Therefore

$$\mathcal{W}_2(\tilde{\mathbb{P}}_{T_k}, \mu_{\boldsymbol{D}, T_k}) \leq \sqrt{C_6 T_k \sum_{j=1}^{k-1} \epsilon_{j+1}^2}.$$

Then we have

$$\mathcal{W}_2(\nu_{\boldsymbol{D},T_k}, \mu_{\boldsymbol{D},T_k}) \leq \mathcal{W}_2(\tilde{\mathbb{P}}_{T_k}, \nu_{\boldsymbol{D},T_k}) + \mathcal{W}_2(\tilde{\mathbb{P}}_{T_k}, \mu_{\boldsymbol{D},T_k})$$

$$\leq \sqrt{C_5 \log(T_k)} \left( \sqrt{\tilde{C}(k)} + \left( \frac{\tilde{C}(k)}{2} \right)^{1/4} \right) + \sqrt{C_6 T_k \sum_{j=1}^{k-1} \epsilon_{j+1}^2}.$$

$\square$

**Remark S2** *The constant $\varsigma$ in (S22) comes from Assumption B4, which controls the difference between $\nabla_{\boldsymbol{Z}} \hat{F}_{\boldsymbol{D}}(\boldsymbol{Z}, \boldsymbol{\theta})$ and $\nabla_{\boldsymbol{Z}} F_{\boldsymbol{D}}(\boldsymbol{Z}, \boldsymbol{\theta})$. When the full data is used at each iteration of Algorithm 1, $\nabla_{\boldsymbol{Z}} \hat{F}_{\boldsymbol{D}}(\boldsymbol{Z}, \boldsymbol{\theta}) = \nabla_{\boldsymbol{Z}} F_{\boldsymbol{D}}(\boldsymbol{Z}, \boldsymbol{\theta})$ and thus the term $C_3 \varsigma T_k$ disappears. In this case, for any fixed time $T_k = t$ and for any decaying sequences $\{\epsilon_k\}$ and $\{\gamma_k\}$, we have $\sum_{j=0}^{k-1} \epsilon_{j+1}^2 \leq T_k \epsilon_1$ and $\sum_{j=0}^{k-1} \epsilon_{j+1} \gamma_j \leq T_k \gamma_1$. Therefore, we can make $\mathcal{W}_2(\nu_{\boldsymbol{D},T_k}, \mu_{\boldsymbol{D},T_k})$ arbitrarily small by setting smaller values of $\epsilon_1$ and $\gamma_1$.*

The convergence of $\nu_{\boldsymbol{D},T_k}$ to its stationary distribution can be quantified by Theorem 19 of Gao et al. (2021):

**Lemma S2 (Gao et al. (2021))** *Suppose Assumptions B1-B7 hold. Then there exist constants $C$ and $\mu^*$ such that $\mathcal{W}_2(\nu_{\boldsymbol{D},T_k}, \pi_{\boldsymbol{D}}) \leq C \sqrt{\mathcal{H}_\rho(\mu_0, \pi_{\boldsymbol{D}})} e^{-\mu_* T_k}$, where $\mathcal{H}_\rho$ is a semi-metric for probability distributions, and $\mathcal{H}_\rho(\mu_0, \pi_{\boldsymbol{D}})$ measures the initialization error.*

Please refer to Theorem 19 in Gao et al. (2021) for more details about the constant $C$ and $\mathcal{H}_\rho(\mu_0, \pi_{\boldsymbol{D}})$. Together, we have

$$\mathcal{W}_2(\mu_{\boldsymbol{D},T_k}, \pi_{\boldsymbol{D}}) \leq \mathcal{W}_2(\mu_{\boldsymbol{D},T_k}, \nu_{\boldsymbol{D},T_k}) + \mathcal{W}_2(\nu_{\boldsymbol{D},T_k}, \pi_{\boldsymbol{D}})$$

$$\leq C \sqrt{\mathcal{H}_\rho(\mu_0, \pi_{\boldsymbol{D}})} e^{-\mu_* T_k} + \sqrt{C_5 \log(T_k)} \left( \sqrt{\tilde{C}(k)} + \left( \frac{\tilde{C}(k)}{2} \right)^{1/4} \right) + \sqrt{C_6 T_k \sum_{j=1}^{k-1} \epsilon_{j+1}^2}, \quad \text{(S30)}$$

which can be made arbitrarily small by choosing a large enough value of $T_k$ and small enough values of $\epsilon_1$ and $\gamma_1$, provided that $\{\epsilon_k\}$ and $\{\gamma_k\}$ are set as in Theorem S1. This completes the proof of Theorem 3.2.

## 4 Parameter Settings Used in Numerical Experiments

For all these datasets, we use $n$ to denote the sample size of the training set.

### 4.1 Binary Classification Examples

**thyroid** The StoNet consisted of one hidden layers with $q$ hidden units, where $ReLU$ was used as the activation function, $\sigma_{n,1}^2$ was set as $10^{-7}$, and $\sigma_{n,2}^2$ was set as $10^{-9}$. For HMC imputation, $t_{HMC} = 25$, $\eta = 100$. In the $\boldsymbol{\theta}$-training stage, we set the mini-batch size as 64 and trained the model for 500 epochs, $\gamma_{k,1} = (3e-5)/n$ and $\epsilon_k = 0.001$ for all $k$. In the SDR stage, we trained the model with the whole dataset for 30 epochs. Besides, the learning rate $\epsilon_k$ was set as $\frac{1}{1000+k^{0.6}}$ and the step size $\gamma_{k,1}$ was set as $\frac{1/n}{1/(3e-5)+k^{0.6}}$.

**breastcancer** The StoNet consisted of one hidden layers with $q$ hidden units, where $ReLU$ was used as the activation function, $\sigma_{n,1}^2$ was set as $10^{-5}$, and $\sigma_{n,2}^2$ was set as $10^{-6}$. For HMC imputation, $t_{HMC} = 25$, $\eta = 100$. In the $\boldsymbol{\theta}$-training stage, we set the mini-batch size as 32 and trained the model for 100 epochs, $\gamma_{k,1} = (1e-4)/n$ and $\epsilon_k = 0.001$ for all $k$. In the SDR stage, we trained the model with the whole dataset for 30 epochs. Besides, the learning rate $\epsilon_k$ was set as $\frac{1}{1000+k^{0.6}}$ and the step size $\gamma_{k,1}$ was set as $\frac{1/n}{10000+k^{0.6}}$.

**flaresolar** The StoNet consisted of one hidden layers with $q$ hidden units, where $ReLU$ was used as the activation function, $\sigma_{n,1}^2$ was set as $10^{-5}$, and $\sigma_{n,2}^2$ was set as $10^{-6}$. For HMC imputation,

$t_{HMC} = 25$, $\eta = 100$. In the $\boldsymbol{\theta}$-training stage, we set the mini-batch size as 32 and trained the model for 100 epochs, $\gamma_{k,1} = (7e-5)/n$ and $\epsilon_k = 0.001$ for all $k$. In the SDR stage, we trained the model with the whole dataset for 30 epochs. Besides, the learning rate $\epsilon_k$ was set as $\frac{1}{1000+k^{0.6}}$ and the step size $\gamma_{k,1}$ was set as $\frac{1/n}{1/(7e-5)+k^{0.6}}$.

**heart, german** The StoNet consisted of one hidden layers with $q$ hidden units, where $Tanh$ was used as the activation function, $\sigma_{n,1}^2$ was set as $10^{-7}$, and $\sigma_{n,2}^2$ was set as $10^{-8}$. For HMC imputation, $t_{HMC} = 25$, $\eta = 100$. In the $\boldsymbol{\theta}$-training stage, we set the mini-batch size as 64 and trained the model for 100 epochs, $\gamma_{k,1} = (5e-5)/n$ and $\epsilon_k = 0.001$ for all $k$. In the SDR stage, we trained the model with the whole dataset for 30 epochs. Besides, the learning rate $\epsilon_k$ was set as $\frac{1}{1000+k^{0.6}}$ and the step size $\gamma_{k,1}$ was set as $\frac{1/n}{20000+k^{0.6}}$.

**waveform** The StoNet consisted of one hidden layers with $q$ hidden units, where $ReLU$ was used as the activation function, $\sigma_{n,1}^2$ was set as $10^{-3}$, and $\sigma_{n,2}^2$ was set as $10^{-6}$. For HMC imputation, $t_{HMC} = 25$, $\eta = 10$. In the $\boldsymbol{\theta}$-training stage, we set the mini-batch size as 64 and trained the model for 30 epochs, $\gamma_{k,1} = (7e-4)/n$ and $\epsilon_k = 0.01$ for all $k$. In the SDR stage, we trained the model with the whole dataset for 30 epochs. Besides, the learning rate $\epsilon_k$ was set as $\frac{1}{1000+k^{0.6}}$ and the step size $\gamma_{k,1}$ was set as $\frac{1/n}{1/(7e-4)+k^{0.6}}$.

We used the module $LogisticRegression$ of $sklearn$ in Python to fit the logistic model.

## 4.2 Multi-label Classification Example

**Hyperparameter settings for the StoNet** The StoNet consisted of one hidden layers with $q$ hidden units, where $Tanh$ was used as the activation function, $\sigma_{n,1}^2$ was set as $10^{-3}$, and $\sigma_{n,2}^2$ was set as $10^{-6}$. For HMC imputation, $t_{HMC} = 25$, $\eta = 10$. In the $\boldsymbol{\theta}$-training stage, we set the mini-batch size as 128 and trained the model for 20 epochs, $\gamma_{k,1} = (7e-4)/n$ and $\epsilon_k = 0.01$ for all $k$. In the SDR stage, we trained the model with the whole dataset for 30 epochs. Besides, the learning rate $\epsilon_k$ was set as $\frac{1}{100+k^{0.6}}$ and the step size $\gamma_{k,1}$ was set as $\frac{1/n}{1/(7e-4)+k^{0.6}}$.

**Hyperparameter settings for the autoencoder** We trained autoencoders with 3 hidden layers and with $400, q, 400$ hidden units, respectively. We set the mini-batch size as 128 and trained the autoencoder for 20 epochs. $Tanh$ was used as the activation function and the learning rate was set to 0.001.

**Hyperparameter settings for the neural network** We trained a feed-forward neural network on the dimension reduction data for the multi-label classification task and another neural network on the original dataset as a comparison baseline. The two neural networks have the same structure, one hidden layer with 50 hidden units, and have the same hyperparameter settings. We set the mini-batch size as 128 and trained the neural network for 300 epochs. $Tanh$ was used as the activation function and the learning rate was set to 0.01.

## 4.3 Regression Example

**Hyperparameter settings for the StoNet** The StoNet consisted of 2 hidden layers with 200 and $q$ hidden units, respectively. $Tanh$ was used as the activation function, $\sigma_{n,1}^2$ was set as $10^{-5}$, $\sigma_{n,2}^2$ was set as $10^{-7}$, and $\sigma_{n,3}^2$ was set as $10^{-9}$. For HMC imputation, $t_{HMC} = 25$, $\eta = 10$. In the $\boldsymbol{\theta}$-training stage, we set the mini-batch size as 800 and trained the model for 500 epochs, set $\gamma_{k,1} = (7e-5)/n$, $\gamma_{k,2} = (7e-6)/n$ and $\epsilon_k = 0.01$ for all $k$. In the SDR stage, we trained the model with the whole dataset for 30 epochs. Besides, the learning rate $\epsilon_k$ was set as $\frac{1}{100+k^{0.6}}$, the step size $\gamma_{k,1}$ was set as $\frac{1/n}{1/(7e-5)+k^{0.6}}$, and $\gamma_{k,2}$ was set as $\frac{1/n}{1/(7e-6)+k^{0.6}}$.

**Hyperparameter settings for the autoencoder** We trained autoencoders with 3 hidden layers and with $200, q, 200$ hidden units, respectively. We set the mini-batch size as 800 and trained the neural network for 20 epochs. $Tanh$ was used as the activation function and the learning rate was set to 0.01.

**Hyperparameter settings for the neural network**   We trained a feed-forward neural network on the dimension reduction data for making predictions and another neural network on the original dataset as a comparison baseline. The two neural networks have the same structure, one hidden layer with 100 hidden units, and have the same hyperparameter settings. We set the mini-batch size as 32 and trained the neural network for 300 epochs. *Tanh* was used as the activation function and the learning rate was set to 0.03.