# OpenReview forum: "Nonlinear Sufficient Dimension Reduction with a Stochastic Neural Network"
_NeurIPS.cc/2022/Conference — NeurIPS 2022 Accept_

### Official Review · Reviewer_yby7 · 2022-07-11

**Rating:** 6
**Confidence:** 3
**Soundness:** 3 good
**Presentation:** 3 good
**Contribution:** 3 good

**Summary:**

This paper addresses nonlinear sufficient dimension reduction problems using a novel neural network architecture. The key idea is a layer-wise Markovian assumption which ensures the latent representation is identifiable---an issue associated with several related approaches. The authors further provided a stochastic gradient MCMC learning algorithm as well as theoretical analysis of convergence.


**Questions:**

See the main comments.

**Strengths And Weaknesses:**

[Strengths]
1. The paper is clearly written. The idea of the layer-wise Markovian assumption is interesting and natural in terms of the SDR assumption.
2.  Theoretical analysis is provided which shows the proposed StoNet has asymptotically the same loss function as the desired DNN when the training sample size becomes large.
3. Numerical results demonstrate the efficacy of the proposed algorithm in classification and regression.

[Weaknesses]
1. It is not clear how are the depth h and layer width decided in the decomposition Eq (4). How to avoid learning trivial relations such as identity in the components?
2. The StoNet could be easily overparameterized where noise is also captured. Thus, the learned representation could be more than sufficient. Some form of regularization could be needed.
3. The misclassification rate for GSAVE and GSIR seems unrealistically high compared to the linear methods. Is there intuition why this is the case?

---

> ### Author Response · Authors · 2022-08-02
> **On StoNet structure**
>
> 1. In our paper, we study SDR with a given network structure, under the assumption that the network structure has been large enough for approximating the underlying true nonlinear function. To determine the optimal depth, layer width, etc, we can combine our method with a sparse deep learning method e.g. [1], [2]. That is, we can start with an over-parametrized neural network, employ a sparse deep learning technique to learn the network structure, and then employ our method to the learned sparse DNN for the SDR task. We particularly note that the work [2] ensures consistency of the learned sparse DNN structure, which can effectively avoid learning trivial relations such as identity in the components.  This point will be added to the revision.
>
> 2.  The over sufficiency issue can be easily addressed by adding a post sparsification step, i.e., applying a sparse SDR procedure (e.g., [3] and [4], working with a Lasso penalty) to the output layer regression of the learned StoNet by noting that the regression at each layer of the StoNet forms a multiple index model. In this way, the StoNet provides a convenient method for determining the central space of SDR for general nonlinear regression model, given its universal approximation ability. This is a remarkable result and will be added to the revision.
>
> 3. It is know that the sliced inverse regression based methods, including SIR and SAVE and their generalized versions, often perform poorly for classification problems as they rely much on $Var(E(X|Y))$ or $Var(X|Y)$, where $Y$ represents class label and only takes very few values. Here $X$ can be interpreted as the original or transformed explanatory variables.  Also, as pointed out by [5], a nonlinear SDR method can sometimes present differences in variances of different classes as a separation of locations in nonlinear sufficient predictor, leading to poor class prediction accuracy.
>
>  ## Reference
> [1] Song Han, Jeff Pool, John Tran, and William Dally.  Learning both weights and connections for efficient neural network.   In Advances in neural information processing systems, pages 1135–1143, 2015.
>
> [2] Y. Sun, Q. Song, and F. Liang. Consistent sparse deep learning: Theory and computation. Journal of the American Statistical Association, page in press, 2021.
>
> [3] Q. Lin, Z. Zhao, and J.S. Liu. On consistency and sparsity for sliced inversion regression in high dimensions. The Annals of Statistics, pages 580-610, 2018.
>
> [4] Q. Lin, Z. Zhao, and J.S. Liu. Sparse sliced inverse regression via Lasso. Journal of the American Statistical Association, pages 1726-1739, 2019.
>
> [5] Li, B., Artemiou, A. and Li, L., 2011. Principal support vector machines for linear and nonlinear sufficient dimension reduction. The Annals of Statistics, 39(6), pp.3182-3210.

---

> > ### Author Response · Authors · 2022-08-08
> > **Follow-up Questions?**
> >
> > Do you have any follow-up questions on our response?  We are looking forward to your further comments.

---

### Official Review · Reviewer_RiM6 · 2022-07-12

**Rating:** 4
**Confidence:** 4
**Soundness:** 3 good
**Presentation:** 3 good
**Contribution:** 2 fair

**Summary:**

The authors propose a nonlinear sufficient dimension reduction method to extract key information in high-dimensional data. They proposed a stochastic neural network and develop an adaptive stochastic gradient Markov chain Monte Carlo algorithm with a convergence theorem established. Extensive empirical experiments on benchmark and real datasets are run to showcase the good prediction performance of the algorithm.


**Questions:**

1. How is the author's model different from [1] Sufficient Dimension Reduction with Deep Neural Networks for Phenotype Prediction by Siqi Liang, Wei-Heng Huang, Faming Liang. Proceedings of the 3rd International Conference on Statistics: Theory and Applications (ICSTA'21)
I am aware that in [1], a divide-and-conquer DNN is used, except this, is there any difference in the model part? To me, the model in the current paper seems not novel compared with the model in [1].

2.Any empirical results to show the computational cost for the author's algorithm and other competing algorithms? It seems to me that the stochastic gradience HMC could be slow for large-scale dnn task even if a mini-batch is used.

3. How is the author's approach compared with [1] as they are so similar? Is there an apparent advantage for the author's method compared with [1]. Why [1] is not cited or compared as a competing algorithm?

**Limitations:**

The authors do not discuss the limitations of their work.

**Strengths And Weaknesses:**

Strengths:
1. The authors develop an adaptive stochastic gradient MCMC algorithm for training the StoNet and provide a rigorous study for its convergence under mild conditions.
2. The authors have provided extensive empirical results to demonstrate the advantage of their method.
Weaknesses:
1. The model for this model seems not novel and I do not find a clear difference between the author's model and the model in the following paper:
[1] Sufficient Dimension Reduction with Deep Neural Networks for Phenotype Prediction by Siqi Liang, Wei-Heng Huang, Faming Liang. Proceedings of the 3rd International Conference on Statistics: Theory and Applications (ICSTA'21)
2. Computational complexity or empirical computational cost analysis and results are not provided so I do not have a clue if the author's algorithm will be much slower than state-of-the-art algorithms in terms of inference.
3. To me, the closest model to the author's work is [1] but no comparison is provided with [1].

---

> ### Author Response · Authors · 2022-08-01
> **Comparison with an existing work**
>
> 1. The work [1] employs a regular DNN for sufficient dimension reduction. As stated in [1], it works only for the case that the output variable is distributed according to a distribution in exponential family. Moreover, the validity of the method in [1] depends on the consistency of the learned DNN model, but [1] does not provide a theoretical guarantee for that.
>
> In this work, we develop a stochastic neural network model with a clearly specified Markov structure such that the proposed method essentially works for any output variable distributions. In addition, we provide a rigorous theoretical guarantee for the validity of the proposed method.
>
> 2. In table 2 of the main paper, we report the running time of different methods. For our Adaptive SGHMC algorithm, the main computational cost comes from evaluations of the gradients. The backward imputation and parameter update steps shares similar computational cost as the standard back propagation algorithm. Hence, the computational cost of each iteration is roughly $t_{HMC} + 1$ times the cost of standard back propagation, which shares the same order as that of SGD. In experiments, we often set $t_{HMC}$ in an order of ten, e.g., 5\~ 25. In theory, it can be as low as 1.
>
> 3. As discussed in point 1, we provided a more rigorous theoretical framework for conducting SDR with neural networks.  In the revision, the work [1] will be cited and discussed.
>
> ## Reference
> [1] Liang, Siqi, Wei-Heng Huang, and Faming Liang. "Sufficient Dimension Reduction with Deep Neural Networks for Phenotype Prediction." Proceedings of the 3rd International Conference on Statistics: Theory and Applications (ICSTA’21). Vol. 134. 2021.

---

> > ### Author Response · Authors · 2022-08-08
> > **Follow-up Questions?**
> >
> > Do you have any follow-up questions on our response?  We are looking forward to your further comments.

---

### Official Review · Reviewer_EGCt · 2022-07-17

**Rating:** 5
**Confidence:** 3
**Soundness:** 2 fair
**Presentation:** 3 good
**Contribution:** 3 good

**Summary:**

This work contributes a few interesting ideas for deep sufficient dimension reduction (SDR). Authors developed a new stochastic neural network and proved that the output of last hidden layer satisfies the SDR condition. Authors also discussed carefully the optimization.

**Questions:**

please see above about weakness

**Limitations:**

It seems authors did not discuss the potential negative social impact.

**Strengths And Weaknesses:**

Using stochastic neural network for SDR is new to me, which is also different from existing deep SDR methods.

My concerns mainly come from the evaluation section. Authors discussed a few issues of exisiting deep SDR methods ([2,20]). However, there is no comparision to any deep methods. Moreover, it is also a little unclear to me how [2] or [20] suffers from these limitations, and how the new method avoids such issue. Could you also empirically valiate this motivation? Additionally, it would be better if authors can compare with kernel dimension reduction.

---

> ### Author Response · Authors · 2022-08-01
> **Non-identifiability issue of deep SDR and Results of Kernel Dimension Reduction**
>
> 1.  We will add an example to demonstrate the non-identifiability issue of the deep SDR method developed [1] and [2].  The outline of the example is given in what follows:
>
> We consider the "M1" data set used in [1], which  consists of 100 training instances of 20 dimensions. The data set is generated by the model $Y = \cos(X b)$, where $X \in R^{100 \times 20}$ is the input data matrix, $B \in R^{20 \times 1}$ is vector with first 6 dimensions equal to $\frac{1}{\sqrt{6}}$ and the other dimensions equal to 0.  We use the code provided by [1] (at https://git.art-ist.cc/daniel/NNSDR/src/branch/master) to conduct the experiment.  We follow the setting in [1], i.e. the deep SDR method reduces the data to a vector (1-dimension) by a refinement network with structure 20-1-512-1. Let $Z_1$ and $Z_2$ denote output vectors obtained by the deep SDR method [1] in two independent runs with different initializations of network weights. We then test the independence of $Z_1$ and $Z_2$ using the R package RCIT, and get a p-value of 0.4068 which does not reject the hypothesis that $Z_1$ and $Z_2$ are independent.
>
> For comparison, we also apply our method to the same data set. We use a StoNet of structure 20-10-1-1 with $\tanh$ as activation function, and use the imputed values of the last hidden layer as the SDR output. In this way, we also reduce the data to a vector (1-dimension). We also conduct two independent runs with different initializations of network weights and use the R package RCIT to test independence of the SDR output vectors. The resulting p-value is 0.012, which suggests that the our SDR output vectors are not independent.
>
> This example implies that the deep SDR method (in [1] and [2]) may suffer from the non-identifiability issue, while ours is not.
>
> 2. As suggested, we have tested the kernel dimension reduction (KDR) method on same examples in Table 1. The results are given below.  The comparison indicates that the proposed method is superior to KDR, while KDR is comparable with other existing methods such as SIR and GSAVE.
>
> | Datasets     | q  | StoNet             | KDR                |
> |--------------:|:----:|:--------------------:|:--------------------:|
> | thyroid      | 1  | **0.0687(.0068)**  | 0.2847 (.0110)     |
> |              | 2  | **0.0693(.0068)**  | 0.2713 (.0128)     |
> | breastcancer | 2  | **0.2578(.0074)**  | **0.2714 (.0102)** |
> |              | 4  | **0.2682(.0113)**  | **0.2740 (.0105)** |
> | flaresolar   | 2  | **0.3236(.0040)**  | 0.4161 (.0138)     |
> |              | 4  | **0.3239 (.0043)** | 0.3673 (.0108)     |
> | heart        | 3  | **0.1625(.0076)**  | 0.1870 (.0064)     |
> |              | 6  | **0.1625(.0062)**  | **0.1715 (.0075)** |
> | german       | 5  | **0.2368(.0050)**  | **0.2430 (.0050)** |
> |              | 10 | **0.2356(.0047)**  | **0.2347 (.0075)** |
> | waveform     | 5  | **0.1091(.0010)**  | 0.1269(.0031)      |
> |              | 10 | **0.1079(.0012)**  | 0.1254(.0030)      |
>
>
> ## Reference
> [1] Daniel Kapla, Lukas Fertl, and Efstathia Bura. Fusing sufficient dimension reduction with neural networks. Computational Statistics & Data Analysis, 2021.  (Reference [20] in the paper)
>
> [2] Ershad Banijamali, Amir-Hossein Karimi, and Ali Ghodsi. Deep variational sufficient dimensionality reduction. In Third Workshop on Bayesian Deep Learning (NeurIPS 2018), 2018. (Reference [2] in the paper)

---

> > ### Author Response · Authors · 2022-08-08
> > **Follow-up Questions?**
> >
> > Do you have any follow-up questions on our response?  We are looking forward to your further comments.

---

### Meta-Review · Area_Chair_HQ8b · 2022-08-25

**Recommendation:** Accept
**Confidence:** Certain

**Metareview:**

This paper gives a new method to perform nonlinear sufficient dimension reduction by utilizing a stochastic neural network. The derivation of the proposed method is justified by some theoretical background, and a convergence rate analysis is given for the derived algorithm. The practical performance is evaluated by some numerical experiments on real datasets.

Although there are some related work, the proposed model is new. The theoretical justification of the proposed method is solid. The paper is overall clearly written.
The numerical experiments properly shows effectiveness of the method (while they are rather small).

In summary, this paper presents a novel method with nice theoretical and numerical justifications. I recommend acceptance.

**Award:**

No

---

### Decision · Program_Chairs · 2022-09-14

Accept